# Fluorinated Protein and Peptide Materials for Biomedical Applications

**DOI:** 10.3390/ph15101201

**Published:** 2022-09-28

**Authors:** Julia M. Monkovic, Halle Gibson, Jonathan W. Sun, Jin Kim Montclare

**Affiliations:** 1Department of Chemical and Biomolecular Engineering, NYU Tandon School of Engineering, Brooklyn, NY 11201, USA; 2Department of Chemistry, New York University, New York, NY 10003, USA; 3Department of Radiology, New York University Langone Health, New York, NY 10016, USA; 4Department of Biomedical Engineering, NYU Tandon School of Engineering, Brooklyn, NY 11201, USA; 5Department of Biomaterials, New York University College of Dentistry, New York, NY 10010, USA

**Keywords:** fluorine, fluorous effect, perfluorocarbons, rational design, non-canonical amino acids, biologics, drug delivery, bioimaging

## Abstract

Fluorination represents one of the most powerful modern design strategies to impart biomacromolecules with unique functionality, empowering them for widespread application in the biomedical realm. However, the properties of fluorinated protein materials remain unpredictable due to the heavy context-dependency of the surrounding atoms influenced by fluorine’s strong electron-withdrawing tendencies. This review aims to discern patterns and elucidate design principles governing the biochemical synthesis and rational installation of fluorine into protein and peptide sequences for diverse biomedical applications. Several case studies are presented to deconvolute the overgeneralized fluorous stabilization effect and critically examine the duplicitous nature of the resultant enhanced chemical and thermostability as it applies to use as biomimetic therapeutics, drug delivery vehicles, and bioimaging modalities.

## 1. Introduction

Heralded as a miracle element, fluorine has earned its reputation as one of the most popular heteroatoms in modern chemical design due to its ability to furnish molecules with desirable physiochemical properties, such as honed target selectivity [1], strengthened binding affinity [2], enriched metabolic stability, and prolonged circulation times [3]. The past decade has witnessed the explosive emergence of fluorinated small molecules and nanomaterials in nearly every major commercial industry market, ranging from blockbuster pharmaceuticals like atorvastatin (Lipitor) to non-stick cookware, liquid crystalline displays to fluorinated anesthetics, refrigerants to toothpaste [4].

Although the impact of a fluorine atom on the physiochemical properties of small molecules and fluoropolymers is well-known, robust understanding of how fluorination contributes to the structural integrity of proteins remains relatively unexplored. Proteins or polypeptides are comprised of amino acid (AA) building blocks that can spontaneously self-assemble into intricate higher-order structures. The manner by which these AA side-chains extend out from the protein backbone and interact with one another dictates the stability and complexity of the protein fold. Named after their characteristic backbone covalent linkages, peptides are like proteins with the key distinction that they often lack an organized secondary structure. Exquisite prediction and control of the protein and peptide self-assembly lies at the crux of protein engineering.

Beyond the fundamental appeal of investigating new fluorinated architectures, fluorinated protein and peptide-based biomaterials hold translational potential as attractive biodegradable alternatives to fluoropolymeric systems [5]. While conventional wisdom in the materials selection process is to seek durable materials that maintain their structural integrity over long durations of time, recent scrutiny of biopersistent polymeric materials constructed from perfluorinated alkyl substances (PFAS) has generated a unique niche for fluorinated proteins and peptides that can perform their intended biorthogonal function and disappear soon after in a safe and efficient manner.

This review begins by examining fundamental properties of the fluorine atom and methods for introducing fluorine into protein and peptide biomolecules. Afterwards, bioisosteric tactics for fluorination derived from small molecule medicinal chemistry are summarized to preface a deconstruction of fluorine’s impact on secondary and supersecondary protein structures. This passage highlights the fluorous stabilization phenomenon by which fluorine’s superhydrophobic nature compels it to associate with similar fluorinated moieties and is contextualized against competing intramolecular forces driving protein self-assembly and ultimately thermodynamic stability. Finally, the application of these functionalized fluorinated materials as active therapeutic ingredients, drug delivery vehicles, and bioimaging modalities in the biomedical space are examined under a critical lens of sustainable best practices.

### Special Properties of Fluorine

Unless otherwise specified, fluorine as it is mentioned in this review refers to fluorine-19 (^19^F). The approximately 99% natural abundance of ^19^F has led to it being regarded as a monoisotopic element [6]. Nevertheless, several other radioisotopes of fluorine ranging from the lightest fluorine-13 to the heaviest fluorine-31 have also been discovered through cyclotron particle acceleration. In general, the short lifetimes of these exotic radionuclide (on the order of femtoseconds) render them insuitable for practical use [7]; however, fluorine-18 (^18^F) with its relatively longer beta decay into stable oxygen-18 (^18^O) over the course of 110 min has been rather extensively studied as a bioimaging radiolabel [8], which will be further discussed in *Fluorine-18 Radiotracers in Positron Emission Tomography Imaging*.

Fluorine primarily draws its reputed inductive benefits from its unique electronic profile. With a ground-state electron configuration of [He] 2s^2^ 2p^5^, fluorine only has two valence electron shells, causing its electrons to cling tightly to the nucleus and drastically reduce the space the atom occupies. Concurrently, this suggests that fluorine’s positively-charged core is poorly shielded, heightening its propensity to seize negatively-charged electrons from atoms in the surrounding environment. Thus, fluorine is the most electronegative element by far with a dimensionless value of 3.98 on the Pauling electronegativity scale [9]. Fluorine’s strong affinity for electrons compels it to form extremely stable, polar covalent bonds with its atomic partners due to the large difference in electronegativity.

Beyond its robust ability to form covalent bonds, the aggressive electron-withdrawing nature of fluorine is also responsible for modulating the electron density of surrounding functional groups. It has been well-established that positioning fluorine adjacent to a protonated moiety (e.g., carboxylic acid, alcohol, amine, etc.) will divert electron density away from the protons via σ-induction and decrease in pKa [10]. Nowhere is this effect more prominantly observed than in the substitution of electropositive hydrogen in aromatic rings. To illustrate, the normal electrostatic profile of non-fluorinated benzene appears highly negative due to the delocalized π-electron cloud and partially positive aryl hydrogen atoms (Figure 1). Conversely, attaching fluorines to benzene’s periphery draws electron density away from the center, effectively inverting the molecule’s net electrostatic profile. The equal but opposite quadrapole moments of benzene and its highly fluorinated aromatic analogs allow the pair of conjugated rings to form stabilizing π–π stacking interactions.

Despite the pronounced polarity of C–F bonds in fluorocarbons, these compounds are counterintuitively hydrophobic in nature. The dense electron clouds surrounding the fluorine atoms protect the molecule from further polarization by fleeting dipoles in solution, resulting in the tendency for fluorocarbons to phase separate in both hydrophilic aqueous and oleophilic organic media. Entropically driven, this phenomena is similar to that of the hydrophobic effect in which highly ordered solvent molecules are displaced by molecule in the fluorous phase which has led to it being regarded as a superhydrophobic effect [10].

## 2. Incorporation of Fluorine into Protein and Peptide Materials

Although it takes the same number of strokes to draw a fluorine in place of a hydrogen atom in a line structure, the actual process of installing fluorine or fluorinated substituents at desired locations in protein or peptide sequences can be quite challenging. Considerable efforts to develop facile fluorination approaches have yielded a plethora of methodologies that are generally classified under two main categories: (1) top-down approaches that introduce fluorine after synthesis of the scaffold and (2) bottom-up approaches that construct fluorinated proteins and peptides using fluorinated AA building blocks (Figure 2). In general, top-down approaches tend to be more time-efficient and economical, but require complex chemistry or specific conditions to execute compared to bottom-up approaches. Depending on the desired level of cellular or chemical infrastructure, these methods may additionally be categorized by the type of techniques employed in the execution of the approach. The following section will provide an overview to some of the powerful engineering tools that facilitate the installation of fluorine into peptides and proteins.

### 2.1. Solid Phase Peptide Synthesis

Culminating from over a century of developments, the efficient chemical synthesis of polypeptide chains marks a major milestone of organic chemistry. Conceived by Dr. R. B. Merrifield who was awarded the 1984 Nobel Prize for his efforts, solid-phase peptide synthesis (SPSS) involves the iterative coupling of AA building blocks onto a solid polymer resin bead one-by-one from the C- to N-terminus direction (Figure 2A) [12]. Although quite labor-intensive, SPSS remains the most frequently employed method since the process is exquisitely controlled through protecting groups (PGs) and offers the possibility of 100% fluorination. In general, this technique cannot generate chains greater than 50 AAs long because as the chain grows, it becomes increasingly likely that unintended cyclization, side-reactions, or chain aggregation will occur. Nevertheless, contemporary synthesizers empowered by automated, flow-based systems continue to improve the technology and proteins of over 100 AAs can now be generated in several hours [13]. Provided that the amine and carboxylic groups remain unaltered, more exotic fluorinated AA analogs including beta-AAs and *N*-substituted glycines can serve as fluorinated building blocks to obtain peptides with fluorinated backbones or poly-*N*-substituted glycines (also called peptoids) as well.

### 2.2. Non-Canonical Amino Acid Incorporation

Non-canonical amino acid (ncAA) incorporation refers to a bottom-up approach that enables co-translational introduction of AA building blocks beyond the twenty conventionally coded for in native host expression systems. First demonstrated by Tirrell and colleagues in an *E. coli* system, AA analogs carrying desired functional groups are introduced to the pre-existing translational machinery, accepted by an endogenous aminoacyl tRNA synthetase (aaRS), and attached covalently to the corresponding tRNA target [14]. This ncAA-tRNA complex proceeds to the ribosome where it can effectively hijack the endogenous cellular circuitry and match an mRNA codon sequence to add the ncAA onto the growing polypeptide chain (Figure 2B). Following this schema, residue-specific installation of ncAA can be achieved globally across a protein by replacing all of one natural AA with its ncAA analog. For this to be effective, the desired ncAA must be present in excess where it can outcompete its canonical counterpart. Approximately 90% incorporation can be reasonably accomplished using an auxotrophic strain of the host cell that cannot make its own AAs and depriving the growth media of the targeted AA [14,15,16]. For instances where global fluorine substitution is not desired, site-specific incorporation of ncAA can be attained by repurposing suppressor tRNAs at amber (AUG) stop codons to append desired ncAA instead of halting protein production [17]. Further nuances and advancements to this powerful synthetic biology technique have been extensively described in other reviews [18,19].

### 2.3. Protein Fragment Ligation

Ligation encompasses a broad class of top-down semisynthetic reactions that merge protein or peptide fragments created by either SPSS or recombinant synthesis in an end-to-end convergent fashion (Figure 2C). Originally, native chemical ligation chemistry could only be performed on sequences with terminal cystines [20]. However, the field of ligation chemistry has dramatically evolved to accommodate fragments with different end residues or functional groups with the advent of Staudinger, α-ketoacid-hydroxylamine, and serine-threonine ligations [21]. In addition to these synthetic chemistry-based ligations, a small subsection of enzymatic ligation strategies utilizing engineered ligases, such as sortase, butelase, peptiligase, or omniligase to catalyze peptide bond formation have been steadily gaining traction as more chemoselective alternatives whose reagents are derived from renewable sources [22]. The principle of ligation methods does not permit modification of residues on the existing fragments, so ligation is typically reserved to install fluorinated endcaps or append fluorinated residues at the ends of sequences. Nevertheless, ligation remains a powerful tool to introduce significant amounts of fluorine in a time-efficient manner and can be used in tandem with the aforementioned bottom-up approaches.

### 2.4. Bulk Protein Fluorination via Fluorinating Agents

Proteins contain many reactive functionalities along their surface that can engage in fluorination reactions. In this top-down approach, proteins are treated as reactants in a fluorination reaction where conventional synthetic chemistry techniques can be employed. A variety of nucleophilic, electrophilic, and even radical fluorinating agents, such as Deoxofluor^®^ and Selectfluor^®^, have been developed to fluorinate specific functional groups [23]. Although these reagents could technically achieve residue-specific fluorination, it would be more accurate to consider this method functional group specific. Indeed, this global fluorination method indiscriminately alters any reactive functionalities that are considered degenerate. Deliberate partial fluorination may be attainable through selective chemical or steric protection of desired moieties, but asymmetric capping of functionalities that are chemically similar is incredibly tricky. Moreover, the use of dry, organic solvents that are often used to promote solubility of fluorinating agents can denature protein structures, further diminishing yield and limiting applicability of this approach [24].

### 2.5. Post-Translational Modifications with Fluorine

Post-translational modifications cover an important class of biological pathways that modify existing proteins with additional chemical functionalities. Leveraging native or chemically modified enzymes to fluorinate proteins at specific sites, this top-down approach offers potential of site-specific resolution via enzymatic recognition of specific peptide sequences (Figure 2D). Direct installation of C–F bonds into post-translational precursors or cofactors has primarily been explored with the enzyme fluorinase. Naturally found in an exotic bacterial strain of *Streptomyces cattleya*, fluorinase participates in a biosynthetic pathway to nucleophilic installation of fluorine onto *S*-adenosine and produce fluoroacetate (FAc) and 4-fluorothreonine (4-FT) downstream of an enzyme cascade (Figure 3) [25]. As the only fluorinated AA found in nature, 4-FT is efficiently incorporated into native *S. cattleya*’s proteins by an amino acyl transfer RNA deacylase and a 4-fluorothreonine transporter [26,27]. Subsequent efforts to decrease the number of steps involved in this enzyme cascade and engineer alternative routes to fluorinate other AAs are currently underway.

Separately, Wang and coworkers have recently demonstrated steric-free installation of fluoroacetamides on substrate proteins by supplementing acetyl transferase with a fluorinated analog of acetyl coenzyme A, effectively hijacking the coenzyme A metabolic pathway [28]. Although their substrate specificity has yet to be fully understood, these post-translational writing enzymes for fluorinating proteins of interest are predicted to play an increasingly prominent role in the foreseeable future.

## 3. Impact of Fluorine on Secondary Structure

Small molecule medicinal chemists often analogize the behavior of fluorine to other substituents as bioisostere pairs that can readily be swapped to enhance certain biophysical properties without making significant changes to a drug candidate’s shape or structure. The small dimensions of fluorine with a van der Waals radius (r_vdW_) of 147 pm have often been compared to hydrogen (r_vdW_ = 120 pm) and oxygen (r_vdW_ = 152 pm), which enable their single atom replacement with minor steric perturbations [29]. The subtle 27 pm difference in radii between fluorine and hydrogen, however, becomes increasingly more conspicuous with additional germinal. To illustrate, the bulkiness of a trifluoromethyl (–CF_3_) group is nearly twice that of a non-fluorinated methyl (–CH_3_) group, making it analogous to an ethyl (–CH_2_CH_3_) group or even an isopropyl group (–CH(CH_3_)_2_) [9]. Due to electronic and steric repulsions of multiple bulky fluorine atoms in proximity with one another, extended fluorocarbon chains have been reported to adopt a slight helical structure [30]. This notion holds interesting implications for perfluorinated moieties since they may cause conformational deviations from idealistic bioisosteric behavior when appended to proteins. Electronically, fluorine substituents have been reported to effectively mimic hydroxyl, amine, carbonyl, nitrile, nitro, and even thiol functionality in small molecules (Figure 4). Most relevant to protein engineering and design, the vinylidene fluoride (C=C–F) motif resembles the N–C(O) peptide bond [31]. Despite their similar electron configurations, fluorine surprisingly does not imitate other halogens like chlorine or bromine well since fluorine lacks halogen bond donor capabilities and its anionic form does not function as a viable leaving group in most situations. Evidently, the circumstances of where fluorine mimics these functionalities are heavily context-dependent with no consistent opinion regarding the stereoelectronic nature of fluorine and how the high electronegativity of this atom inductively affect AA side chains to this day [31].

When translating these bioisostere principles to macromolecules, more fluorine atoms tend to be necessary to achieve the same pronounced effect [10]. In other words, monofluorination of small molecules will not have the same impact as monofluorination of a large protein or peptide. When designing or engineering fluorinated proteins or peptides, understanding the principles governing where this element can be installed, how many fluorines are required to achieve a desired effect, and what will change in the surrounding biochemical environment is key. In fact, understanding of how fluorine impacts secondary structure is critical to protein design and engineering since structure begets function is a key tenet of biology. The following section will focus on specific case studies highlighting how these properties translate to key structural moieties often utilized in protein and peptide therapeutic design (Table 1). Fluorinated analogues of hydrophobic AAs show promise as modulators for stability and self-organization of folding motifs whose interactions are largely based on complementary hydrophobic side-chain packing.

### 3.1. Monomeric Alpha-Helices

The α-helix is the most common secondary structure formed when AAs wind up into a right-handed helix where the side-chains point out from the central coil. An α-helix has 3.6 residues per turn [37]. Typically, the resulting helices are stabilized via hydrogen bonding that occurs between the carbonyl oxygen of one AA and the amide proton of another AA four resides away. Because fluorine has both hydrophobic and lipophobic tendencies, expansive work has been performed to further stabilize α-helices via a fluorous stabilization effect by incorporating fluorine into hydrocarbon chains. Pioneering studies include substituting canonical hydrocarbon AAs with highly fluorinated AA analogs to enhance protein stability, with most efforts replacing the hydrophobic residue leucine (Leu) with a 5,5,5,5′,5′,5′-hexafluoroleucine (Hfl) variant [38,39,40].

Identifying an oversight in these studies, Chiu et al. performed a helix propensity study on KHfl formulated with leucine (Leu) and Hfl to assess whether an energetic or geometric factor beyond the hydrophobic effect could influence the self-assembly of α-helical proteins (Table 1) [32]. Substituting Leu with Hfl was found to decrease the helix propensity by 8-fold, while still enhancing the overall thermostability of resultant helical proteins as determined by circular dichroism (CD) spectroscopy (Figure 5). This somewhat contradictory observation can be rationalized through the notion that the fluorinated side chains, although bulkier and more exposed to solvation, closely preserve the shape of the hydrophobic amino acids they replace. Thus, the fluorous stabilization effect can overcome the less favorable helix propensity of Hfl. Alternative routes of stabilization such as those involving the aligned dipole moments of Hfl’s fluoromethyl groups have been postulated to aid in the increase in helical stability. To fully comprehend the full extent of the fluorous effect in the creation of α-helices, investigators have sought to design highly fluorinated versions of AAs with higher helix propensities. With a helical propensity of 0.26 ± 0.3, (2*S*,3*S*)-5,5,5-trifluoroisoleucine has displayed the highest helical propensity to date; however, this value is still significantly lower than that of non-fluorinated isofluoroleucine (0.52 ± 0.05) [41]. Fluorinated canonical and non-canonical AAs alike have consistently displayed lower helical propensities compared to their non-fluorinated versions likely due to the previously described steric and solvation issues [42].

### 3.2. Beta-Sheets

The second most common secondary structure found in proteins and peptides is the *β*-sheet, which is composed of several *β*-strands held together by stabilizing hydrogen bonds between adjacent strands. In contrast to α-helices, the residues involved in hydrogen bonding are separated from each other by long segments of AAs [43]. In an attempt to improve stability of *β*-sheets, several studies have replaced AA residues involved in these hydrogen bonding interactions with fluorinated analogues, aiming to exploit the electron-withdrawing properties of fluorine to increase the hydrogen bonding strength, yet the findings have provided mixed results [32,33,34]. Horng and Raleigh replaced valine with 4,4,4-trifluorovaline (tfV) in two interior positions of a *β*-sheet region in a ribosomal protein to create tfV3 and tfV21 (Figure 6A, Table 1). Titrating with a common protein denaturant guanidinium hydrochloride, the authors experimentally determined the Gibbs free energy associated with protein unfolding via monitoring with CD spectroscopy (Figure 6B, Table 1). Both substitutions resulted in an increase in unfolding free energy (ΔΔG°_tfV3_ = 0.79 kcal/mol and ΔΔG°_tfV21_ = 1.4 kcal/mol) indicating a slower propensity to unfold and thus a higher thermostability [33]. Comparatively, Kumar and coworkers methodically studied Hfl incorporation at cross-strand positions of two small *β*-hairpin scaffolds. Surprisingly, replacing interior-facing, non-hydrogen bonding positions in scaffold 1 with Hfl to create HH1 exhibited a slight destabilizing effect (ΔΔG°_HH1_ = −0.19 kcal/mol). Furthermore, replacing exterior-facing pairs in scaffold 2 at the N- and C-terminus to create HH2 lowered the folding free energy by approximately −0.4 kcal/mol, indicating the non-fluorinated canonical leucine AA residue may have a higher propensity to self-assemble into β-strands compared to the Hfl [34].

Chiu et al. proposed that the isolated stabilizing effect from fluorination could not be deconvoluted from diagonal cross-strand interactions that have been attributed to the overall twist in *β*-hairpin [44]. To test this, they selected an isolated *β*-sheet segment in the GB1 domain and synthesized GB1-5,5,5′,5′,-tetrafluoroleucine (GB1-Qfl), GB1-Hfl, and GB1-pentafluorophenylalanine (GB1-Pff) [35]. While GB1-QfL and GB1-Hfl are analogs where leucine residues are displaced by non-canonical QfL and Hfl, respectively, GB1-Pff has phenylalanine residues replaced by Pff (Table 1). These parental AAs were chosen given their hydrophobic profiles, allowing them to potentially aid in stabilizing hydrophobic interactions. The group reported a surprising increase in the unfolding free energy associated with each protein (ΔΔG°_GB1-Qfl_ = 0.21 kcal/mol; ΔΔG°_GB1-Hfl_ = 0.29 kcal/mol; ΔΔG°_GB1-Pff_ = 0.34 kcal/mol) [45]. The greatest increase in free energy seen by Pff could be attributed to phenylalanine’s more hydrophobic profile compared to leucine, as more energy is needed to overcome hydrophobic interactions and unfold. Overall, this general increase in stability observed by all AA mutations is surprising given the solvent-exposed positions of the mutation: although fluorine tends to increase hydrophobicity and would therefore theoretically cause solvation issues when introduced on the exterior of a protein, the fluorinated AAs could have a higher *β*-sheet propensity that favors the formation of a *β*-sheet regardless.

### 3.3. Intrinsically Disordered Proteins

Intrinsically disordered proteins (IDPs) refer to a class of proteins that despite lacking a homogenous ordered structure retain a wide array of important biological functions. IDP sequences tend to be short relative to their ordered counterparts, which suggests that chain length is a critical compositional parameter. Several statistic-based composition analysis tools have been developed to evaluate the disorder propensity that predicts the likelihood of specific canonical AAs to promote ordered folding [46]. Following similar design heuristics for generating ordered α-helices, hydrophobic residues like those found in the interior regions of foldable proteins, tend to have lower disorder propensities, whereas polar, charged, and the helix-breaking proline tend to have higher disordered scores.

Based on a previous understanding of the fluorous stabilization effect, fluorinated variants of AAs should theoretically have lower disorder propensities. Pomerantz et al. created a library of peptide oligomers comprised of alternating canonical lysines and *N-ε*-trifluoroacetylated lysines (TFA-lysine) ranging from 5 to 21 residues in length (Table 1) [36]. To dissuade these peptide oligomers from folding upon fluorination, both non-fluorinated and fluorinated residues were employed to induce electrostatic repulsion of the positively-charged residues and thus resist secondary structure formation. CD spectroscopy of the *n* = 7 variant of fluorinated peptide 1 revealed a random-coil structure under aqueous conditions that seemingly undermines the influence of the fluorous stabilization effect. However, when the peptide was measured in an organic solution, the sequences regained its α-helical structure, suggesting that the uncontested hydrogen-bonding capability of the protein backbone overcame the disorder-promoting repulsive forces (Figure 7). This study suggests that the fluorine’s affinity to other fluorinated moieties may not serve as a strong enough driving force to generate order from disorder.

In general, fluorination of non-solvent exposed residues on the hydrophobic interior of proteins will aid in the preservation and stabilization of secondary structure; however, the extent of the fluorous effect’s influence on secondary structure has yet to be fully explored. Studies examining bioisostere replacement of fluorine in exterior hydrophilic residues for its electronic properties are confusing. Moreover, proteins with simultaneous fluorination of both interior and exterior positions have yet to be reported, likely due to issues with protein solubility. Overall, the addition of fluorine does not appear to be a significant thermodynamic driving force promoting protein folding compared to hydrophobic collapse, steric constraints, or solvation effects, but may provide interesting strategic additions to pre-existing, well-designed constructs.

## 4. Impact of Fluorine on Supersecondary Structure

Supersecondary structures are intermediates between the less specific regularity of secondary structures and the highly specific folding of tertiary structures; they result from packing of adjacent secondary structure motifs, namely the α-helix and *β*-sheet [47]. Examples of these higher-order folding patterns include tightly wound α-helices to form coiled-coils and collagen [37], an array of *β*-strands to form a beta barrel [48], and combinations of multiple secondary building blocks to form protein block copolymers [49]. Although they are all assembled from the same secondary domains, these higher-ordered architectures allow proteins and peptides to perform a more diverse set of jobs compared to their individual components. This includes complex enzymatic manipulation, hydrophobic drug binding, nucleic acid condensation, and adhesion [50]. With proper design considerations, these supersecondary structures can form spontaneously. The self-assembling propensity of these higher-order motifs rely on certain driving forces such as hydrophobic interactions, Van der Waals forces, electrostatic interactions, and hydrogen bonds between side chains [47]. Because of the hydrophobic tendencies of fluorine, as well as its polarity and participation in hydrogen bonding, fluorinated AAs have been postulated to be effective in stabilizing supersecondary protein structures as well (Table 2) [51,52]. The following sections will assess the ability of fluorine to do so in a variety of supersecondary motifs.

### 4.1. Coiled-Coils

Coiled-coils typically consist of two to seven right-handed α-helices that are wound around each other forming a left-handed superhelix [37]. The primary structure is amphiphilic and can be characterized by a heptad repeat (*abcdefg*)_n_. Positions *a* and *d* are typically occupied by non-polar residues that form a hydrophobic core at the interface of the helices. By contrast, the positions *e* and *g* are frequently occupied by complementary charged AAs that form interhelical electrostatic interactions. The remaining heptad repeat positions *b*, *c*, and *f* are exposed to the solvent and can be occupied by any hydrophilic residue in principle. Several prior studies had demonstrated that global substitution of the hydrophobic core with highly fluorinated analogues of hydrophobic AAs leads to increased thermal stability [42]. This additional stability to the already stable motif, together with the presence of large internal cavities, has led to intensive research utilizing fluorine’s hydrophobic effect to further exploit the hydrophobic core of coiled-coils for therapeutic and drug delivery applications. 

The large number of leucine residues available in the hydrophobic core of many coiled-coil proteins has inspired the work of mutating this residue with its fluorinated analogue in hopes of incorporating additive stabilizing effects. Tirrell and coworkers have systematically studied the impact of incorporating two different fluorinated leucine variations: 5,5,5-trifluoroleucine (Tfl) and Hfl on the thermostability of different coiled-coil protein systems. Global substitution of four Tfls into GCN4-p1′s hydrophobic pore resulted in higher thermal stability (ΔT_m_ = 13 °C) (Table 2) [53]. To further probe the energetic consequences of introducing fluorine on dimer formation, computational models of GCN4-p1 were generated. Hfl-GCN4-p1 formed fewer stabilizing hydrogen bonds and exhibited poorer Van der Waals packing compared to Tfl-GCN4-p1, which is consistent with the relative inert nature of C–F bonds (Table 2).

Due to the presence of non-homologous substituents on the fourth carbon, Tfl also forms R and S diastereomers that can further promote or inhibit stabilizing hydrogen bond and Van der Waals interactions between dimeric pairs. In the specific case of the GCN4-p1 system, computational calculations of asymmetric *S* and *R* Tfl incorporation on opposing sides of the dimer (ΔΔG^BE^_(4*S*,4*R*)_ = 34.12 kcal/mol and ΔΔG^BE^_(4*R*,4*S*)_ = 46.07 kcal/mol) displayed a modest increase in the net dimeric binding energy compared to enantiopure homodimeric systems of entirely R (ΔΔG^BE^_(4*R*,4*R*)_ = 33.06 kcal/mol) or S (ΔΔG^BE^_(4*S*,4*S*)_ = 28.67 kcal/mol) Tfl stereoisomers (Table 2) [53]. Similar trends regarding the impact of stereochemistry on dimeric coiled-coil thermostability were observed experimentally by Tirrell et al. in a separate system protein A1. For A1, incorporation of eight trifluoroleucines (TFL) [54] or hexafluoroleucines (HFL) [55] resulted in higher increase in thermal stability, demonstrating that the fluorous stabilization effect increases alongside the number of substituted residues (Table 2).

Further work from the Montclare group has revealed that fluorinated coiled-coil fibers via global TFL substitution are more resistant to thermal and chemical denaturation compared to their non-fluorinated counterparts [56]. Such fluorination was achieved biosynthetically by residue-specific incorporation of TFL into two coiled-coil engineered proteins, C and Q, both derived from the homopentameric Cartilage Oligomeric Matrix Protein coiled-coil (COMPcc) domain [56]. In their native state, non-fluorinated C and Q are both unable to form fibers; however, C+TFL and Q+TFL demonstrate higher-order coiled-coil fibril assembly that have been observed via transmission electron microscopy (TEM) (Figure 8, Table 2). These results agree with similar fluorination studies performed by the Tirrell and Marsh labs on a coiled-coil homodimer and an anti-parallel tetramer, respectively [5,40]. By incorporating multiple fluorinated leucine residues into the hydrophobic core, both groups witnessed minimal structural perturbation under thermal conditions that would otherwise denature the wild-type analogs.

Kumar and coworkers also demonstrated similar results installing tfV in addition to TFL in the hydrophobic core of their small coiled-coil homodimer Peptide 2 (Table 2) [57]. Notably, all groups employed bottom-up approaches to fluorinate these higher-order protein structures in a controlled manner. Thus, incorporating fluorinated hydrophobic residues at the *a* and *d* sites of the coiled-coil heptad can therefore increase the buried hydrophobic surface area to promote hydrophobic collapse and coiled-coil self-assembly [52]. These design trends hold consistent across all studies and appear independent of coiled-coil oligomeric state. This drastic decrease in helix propensity upon introducing fluorine atoms could be due to the fluorine side chains being more exposed in the monomeric helix state; however, these side chains may be partially or fully buried in higher-order α-helical assemblies.

### 4.2. Collagen Triple-Helices

Aside from coiled-coils, collagen represents the other basic multi-strand protein motif. Collagen refers to three parallel polypeptide strands in a left-handed, polyproline II-type (PPII) helical conformation that coil about each other with a one-residue stagger to form a right-handed triple helix [61]. The tight packing of PPII helices within the triple helix mandates that every third residue be glycine (Gly), resulting in a repeating XaaYaaGly sequence, where Xaa and Yaa can be any AA. The most common residues in these positions are stereoisomers of proline (Pro) and hydroxyproline (Hyp), respectively, as they form stabilizing dihedral angles that aid in the formation of the triple helix [61]. Collagen’s unique triple helix structure bestows upon it an exceptional mechanical resistance to tensile forces and a unique ability to build extracellular matrices via their non-extensible, ‘rope-like’ peptide backbones. However, collagen has not seen as much experimental fluorination design in recent years; this is because their tight helical packing makes it difficult to accommodate fluorine mutations due to the large size their side chains will impose on the internal dihedral interactions. Replacing Hyp with 4-fluoroproline (flp) in (flpProGly)_7_ has shown to dramatically increase collagen’s conformational stability; this enhanced stability is the result of the increased tendency of flp residues to adopt a C^y^-*exo* or C^y^-*endo* pyrrolidine ring pucker (2*S,*4*R* or 2*S*,4*S* diastereomer, respectively) (Table 2) [62]. These conformational preferences arise from two stereoelectronic effects: a gauche effect that fixes the pyrrolidine ring pucker, and an *n* → π* interaction that stabilizes the *trans* peptide bond [62]. As an example, Raines et al. have demonstrated that a (2*S*,4*S*)-4-fluoroproline (flp) residue is greatly stabilizing in the Xaa position of a collagen model, yet destabilizing in the Yaa position (Figure 9A,B) [63]. By contrast, a (2*S*,4*R*)-4-fluoroproline (Flp) residue was shown to be greatly destabilizing in the Xaa position but stabilizing in the Yaa position. The dichotomous effect of the diastereomers appears to arise from a gauche effect, which alters pyrrolidine ring pucker and hence properly (or improperly) preorganizes main-chain dihedral angles.

To avoid perturbing the internal structure, Cejas et al. have attached L-pentafluorophenylalanine (F_5_-Phe) and native phenylalanine to the N- and C-termini, respectively, of a (GlyProHyp)_10_ collagen model peptide (Table 2) [58]. Doing so allowed them to induce electrostatic π-π stacking interactions between phenylalanine and F_5_-Phe, which propagated end-to-end fibril stacking similar to that of native collagen. The collagen peptide was found to possess a stable triple-helical character and form micrometer-length fibrillar material that resembles collagen fibrils, in addition to having a significantly higher binding energy than that of the non-fluorinated phenylalanine analogue.

Compared to other protein designs, fluorinated collagen design appears to be relatively limited in scope since the triple helix must maintain its dihedral interactions in this three-residue repeat to keep its collagen triple-helix supersecondary structure. Fluorinated endcaps can still be appended to the end termini of collagen mimetic peptides to induce native collagen lateral bundling, which could be useful as a therapeutic for fibrotic diseases and wounds.

### 4.3. Beta Barrels

The beta barrel consists of an array of *β*-strands arranged in an antiparallel manner, with each individual *β*-strand connected to its neighbor through hydrogen bonding. Hydrogen bonds also connect the first and last strands to form a barrel motif with interior and exterior components. This motif is frequently found in proteins whose function is to bind and transport hydrophobic ligands, as their interior region forms a cavity to serve as the binding region [64]. Despite the useful functionality of proteins with this structure, the knowledge of how fluorination impacts *β*-barrel structure and stability is limited.

Welte et al., investigated the effects of single fluorine labelling on phenylalanine and tryptophan residues in Cold shock protein B from *Bacillus subtilis* (*BsCspB*), a relatively small protein that folds into five *β*-strands forming a *β*-barrel structure (Figure 10) [59]**.** Tryptophan residues were fluorinated at the 4-, 5-, and 6-carbon positions of the indole ring (4-^19^F, 5-^19^F, 6-^19^F), whereas phenylalanine labels were fluorinated at the ortho, meta, and para positions of the aromatic ring (2-^19^F, 3-^19^F, 4-^19^F) (Table 2). Thermodynamic and kinetic parameters were systematically measured for each of the six protein variants to quantify the impact of monofluorination on protein folding. To determine thermodynamic stability, free energy against unfolding, (ΔG°), and melting temperature, (T_M_), were measured; the results suggested that there was not significant variation in thermostability with the most stable variant deviating in ΔG° by only 1.6 kJ/mol from the wild type and the melting temperatures of the fluorinated variants (315.6 K < T_M_ < 320.8 K) being close in value to that of the wild type (T_M_ = 316.8 K) [59]**.** Kinetic stability was also shown to be relatively consistent as the kinetic rate constants for unfolding of the fluorine-labelled variants ranged from 31 s^−1^ to 57 s^−1^ and that of the wild type was 40 ± 1 s^−1^ [59]**.** To further assess potential structural changes of the *β*-barrel due to fluorination, the atomic structures of two variants were determined; the structures showed near perfect overlay with wild type *BsCspB* corroborating the prior results, which suggested minimal perturbations to thermodynamic and kinetic stability (Figure 10).

Overall, the fluorine labels on phenylalanine and tryptophan residues did not have a significant impact on thermodynamic stability nor folding kinetics compared to the wild type; additionally, differences in the number and location of the fluorinated residues did not lead to significant variation. This suggested that monofluorinating hydrophobic aromatic residues within a *β*-barrel structure did not lead to significant stabilization of the protein. However, considering the previous observations for *β*-sheets, higher levels of fluorination, such as incorporation of trifluorinated residues, and more compact placement of these fluorinated residues could potentially result in improved kinetic and thermostability of the *β*-barrel.

### 4.4. Protein Block Copolymers

Protein block copolymers are a special type of polymer composed of two or more chemically-distinct AA sequences, known as blocks, which are covalently linked together. Recent engineering advances have allowed for the synthesis of protein-based block copolymers that can provide unique physiochemical and biological properties based on their chemistry and molecular weights [49]. In comparison to synthetic block copolymers, protein block copolymers exploit the tendency of peptides and proteins to self-assemble and adopt ordered conformations which enables better control over structure formation. By intentionally selecting and positioning AA residues, protein block copolymers can be designed to display desired properties, such as specific secondary structures, hydrophobicity patterns, and mechanical integrity. This observation has led to studies synthesizing and characterizing fluorinated protein block copolymers to demonstrate the impact of fluorination on the polymer’s material properties.

In one such study, the Montclare group investigated three fluorinated protein block copolymers comprised of two self-assembling domains (SADs): elastin (E) and COMPcc [60]. The three original block copolymers (EC, CE, and ECE) have been modified via residue-specific incorporation of *para*-fluorophenylalanine (pFF) to yield pFF-EC, pFF-CE, and pFF-ECE (Figure 11A–C, Table 2). Fluorination modulates secondary structure and *T_t_*, as well as the cooperativity of the transition; however, the findings for each fluorinated block copolymer are different, suggesting dependence on the orientation and number of block subunits. The mechanical behavior of all three polymers with respect to either concentration or temperature responsiveness is altered by fluorination, with the incorporation of pFF resulting in greater elastic character in all constructs. This suggests that fluorination may promote supramolecular association that facilitates elastic network formation in protein block copolymers. The results of this study suggest that fluorination impacts the temperature-dependent secondary structure, supramolecular assembly, and most significantly the mechanical properties of protein block copolymers. This highlights the potential of residue-specific fluorination to modify the bulk material properties, specifically mechanical integrity, to elicit rheological and thermoresponsive behavior in protein block copolymers for biomedical applications.

## 5. Protein and Peptide Mimetic Therapeutics

When used for therapeutic purposes, the role of proteins and peptides can be divided into four groups, depending on their pharmacological activity: (i) to replace a protein that is deficient or abnormal; (ii) to augment an existing signaling pathway; (iii) to provide a novel function or activity to the cell; or (iv) to interfere with the native bioactivity of a molecule or organism [65]. Evidently, proteins can perform highly complex and specific functions, giving them the advantage of minimal drug toxicity from interference with off-target body processes. Protein therapeutics are also less likely to elicit an immune response compared to chemically-synthesized alternatives since most are derived from constructs naturally produced by the human body [66]. Given this unique combination of biologically-favorable metrics, the development of protein and peptide therapeutics has witnessed significant growth, with a surplus of successful case studies in both the clinical and academic realm (Table 3).

Despite these several advantages, the use of native proteins and peptides as therapeutics has often been associated with several drawbacks, such as poor absorption, low stability to proteolytic digestion, and rapid clearance. Protein and peptidomimetics have therefore been developed by modifying native protein and peptides, with the aim of obtaining molecules that are more suitable for clinical development [67]. Much of the expansion of protein and peptidomimetics can be attributed to the engineering of monoclonal antibody therapeutics, whose development has enabled the creation of recombinant protein technology [68]. Since then, this same technology has been extended to engineering other molecular types of therapeutic proteins and peptides, including collagen mimetics and bioactive peptides with antimicrobial and antiviral properties to treat illnesses such as cancer, autoimmune, neurological, and endocrine disorders. Most research today focuses on improving the in vivo behavior of these active pharmaceutical ingredients, with fluorination being one of the most common ways to do so: its concoction of special properties could impart enhanced binding interactions, metabolic stability, and selective reactivities to certain protein therapeutics.

Selective fluorination into protein and peptide-based therapeutics has been shown to enhance these pharmacokinetic and physicochemical properties (i.e., absorption, distribution, metabolism, and excretion) [69]. Absorption and distribution are heavily dependent on the molecule’s balance between lipophilicity and hydrophilicity. Permitted that the fluorinated compound does not crash out, fluorination can facilitate favorable partitioning between aqueous polar solution and less polar hydrophobic binding pockets. The compactness of fluorine is also advantageous from a dimensional standpoint as therapeutic substrates can be custom molded to match the confines of targeted binding regions for enhanced affinity through shape complementarity [31]. Occasionally overlooked, metabolic deactivation and excretion must also be considered in the holistic design of a therapeutic. To facilitate clearance, drugs are often subject to metabolizing enzymes, namely cytochrome P450 monooxygenase present in the liver.

**Table 3 pharmaceuticals-15-01201-t003:** Summary of fluorinated proteins and peptides constructs alongside their accompanying one letter AA sequences in the order they are discussed for their application as active therapeutic ingredients.

Construct Name	Fluorinated Residue(s)	Number of Fluorines	Position of Fluorines	Synthetic Method	(Super) Secondary Structure(s)	Sequence	References
BII1F2	Leucine (L)	6 per L (12 total)	Methyl group, δ-carbon	SPPS	Monomer *α*-helix	TRSSRAGLQFPVGRVHR**LL**RK	[70]
BII5F2	Leucine (L)	6 per L (12 total)	Methyl group, δ-carbon	SPPS	Monomer *α*-helix	RAGLQFPVGRVHR**LL**RK	[70]
BII6F2	Leucine (L)	6 per L (12 total)	Methyl group, δ-carbon	SPPS	Monomer *α*-helix	AGLQFPVGRVHR**LL**RK	[70]
BII10F2	Leucine (L)	6 per L (12 total)	Methyl group, δ-carbon	SPPS	Monomer *α*-helix	FPVGRVHR**LL**RK	[70]
M2F2	Leucine (L)	6 per L (12 total)	Methyl group, δ-carbon	SPPS	Monomer *α*-helix	GIGKF**L**HAAKKFAKAFVAE**L**MNS	[70]
M2F5	Leucine (L)	6 per L (30 total)	Methyl group, δ-carbon	SPPS	Monomer *α*-helix	GIGKF**L**HA**L**KKF**L**KAF**L**AE**L**MNS	[70]
DOPA-Phe(4F)-Phe(4F)-OMe	Phenylalanine (F)	4 per F (8 total)	Aryl ring, all positions	Purchased	Linear Tripeptide	^3OH^Y**FF**	[71]
2ZNX (Fluorinated Antibody)	Tryptophan (W)	1 per W (12 total)	Aryl ring, 5-carbon	Residue-specific ncAA incorporation	Antibody (Combination of *α*-helices and *β*-sheets)	Chain A: DIVLTQSPATLSVTPGNSVSLSCRASQSIGNNLH**W**YQQKSHESPRLLIKYASQSISGIPSRFSGSGSGTDFTLSINSVETEDFGMYFCQQSNS**W**PYTFGGGTKLEITGGGGSGGGGSGGGGSDIQLQESGPSLVKPSQTLSLTCSVTGDSITSDY**W**S**W**IRKFPGNRLEYMGYVSYSGSTYYNPSLKSRISITRDTSKNQYYLDLNSVTTEDTATYYCAN**W**DGDY**W**GQGTLVTVSAAHHHHHH Chain B: DIVLTQSPATLSVTPGNSVSLSCRASQSIGNNLH**W**YQQKSHESPRLLIKYASQSISGIPSRFSGSGSGTDFTLSINSVETEDFGMYFCQQSNS**W**PYTFGGGTKLEITGGGGSGGGGSGGGGSDIQLQESGPSLVKPSQTLSLTCSVTGDSITSDY**W**S**W**IRKFPGNRLEYMGYVSYSGSTYYNPSLKSRISITRDTSKNQYYLDLNSVTTEDTATYYCAN**W**DGDY**W**GQGTLVTVSAAHHHHHH	[72]
(flpHypGly)_7_	Proline (P)	1 per P (7 total)	Pyrrolidine ring, 4′-carbon	SPPS	Collagen monomer	**P**^h^PG**P**^h^PG**P**^h^PG**P**^h^PG**P**^h^PG**P**^h^PG**P**^h^PG	[73]

^3OH^Y is used to denote l-3,4-dihydroxyphenylalanine (l-DOPA) derived from tyrosine. ^h^P is used to denote the non-essential hydroxyproline AA. Fluorinated residues are denoted in bold to indicate their location in the sequence.

### 5.1. Bioactive Peptides

Another trait of protein and peptide materials that distinguish them from synthetic fluoropolymers is their intrinsic biological activities that make them prime candidates for adaptation as active pharmaceutical agents. Bioactive peptides are derived from food proteins and can be absorbed in the gastrointestinal system to employ several health benefits, such as preventing diseases or modulating physiological systems [74]. There is a broad range of functions, depending on the sequence of the bioactive peptides, with most mature research concerning the immune system. They remain inactive while their sequences are kept within the parent protein, which also determines their structure, allowing them to be either linear α-helical peptides or β-sheet peptides. They are activated once released by hydrolysis during food processing and/or during gastrointestinal digestion [74]. When developing fluorinated bioactive peptides with antimicrobial and antiviral functions, it is important to limit fluorine mutations and manage the dosing schedule of the therapeutic to ensure that the toxicity does not become too great and that only the specific microbes intended to be killed are targeted. With respect to antimicrobial peptides specifically, studies have shown that the influence of increased hydrophobicity from limited fluorination at single positions can significantly improve proteolytic resistance as well as antimicrobial activity against both Gram-(+) and Gram-(−) bacteria without increasing toxic effects to mammalian cells [74].

#### 5.1.1. Antimicrobial Bioactive Peptides

The development of bioactive peptides with antimicrobial properties has recently surged in association with dramatic reductions in communicable disease mortality; developing new molecules with different modes of action from that of conventional antibiotics are urgently needed to face the growing resistance to antibiotics. Antimicrobial peptides (AMPs) are the first line of defense against pathogens; as opposed to conventional antibiotics that act on unique intracellular targets, AMPs disrupt the bacterial membrane, making it difficult for the microbe to develop resistance. AMPs are therefore promising for therapeutic applications, yet they pose limitations such as degradation by proteolytic enzymes and moderate antimicrobial activity [75]. Because fluorination can lead to superior chemical and thermal stability in proteins and peptides, recent studies have attempted to improve the functionality of AMPs under the assumption that antimicrobial activity maintains a direct relationship with structural stability. For example, Kumar and colleagues have prepared four fluorinated analogues in the buforin (BII1F2, BII5F2, BII6F2, BII10F2) and two in the magainin series (M2F2, M2F5) via incorporation of 5,5,5,5‘,5‘,5‘-2S-hexafluoroleucine (HFL) at selected sites, which were then analyzed for their antimicrobial activity against both Gram-positive and Gram-negative bacterial strains (Figure 12, Table 3) [70]. The increased α-helical content upon fluorination is related to more pronounced antibacterial action in the case of buforins, with minimal inhibitory concentrations (MICs) around 40 and 10 µg/mL for *E. coli* and *B. subtilis*, respectively; compared to the non-fluorinated version of this same antimicrobial peptide, the MICs for *E. coli* and *B. subtilis* were both over 256 µg/mL [70]. Comparatively, higher amounts of secondary structure correlate with increased hemolysis and formation of aggregates in the magainin series. In looking at the MICs for the same bacterial strains, one fluorinated peptide had values of 40 and 10 µg/mL, compared to the non-fluorinated version which had 2.5 µg/mL for both strains [70]. Therefore, although fluorination may be an effective strategy to increase the stability of biologically active peptides, that does not necessarily guarantee its success as an antimicrobial agent.

#### 5.1.2. Antiviral Bioactive Peptides

In addition to the growing interest in antimicrobial peptides, the outbreak of the COVID-19 global pandemic has motivated researchers to expand the immunological properties of bioactive peptides by developing novel strategies and methods to prevent the spread of viruses. Although some viruses cannot spread outside the body, others, especially bacteriophage and respiratory infection viruses, can easily attach to surfaces and remain adherent, which increases the risk of infection. By providing these surfaces with an antiviral coating that can eliminate the viruses quickly after attachment, the risk of viral transmission can be lowered. Self-assembling peptides are beginning to exhibit potential as candidates for fabricating functional coatings. Reches et al. have demonstrated that a peptide-based assembly with *para*-fluorophenylalanine (Phe(4F)) can kill viruses and form an antiviral coating (Figure 13, Table 3) [71]. The peptide, DOPA-Phe(4F)- Phe(4F)-OMe, forms spherical assemblies with a *β*-sheet like structure, which when cast on a surface, adheres and forms a stable hydrophobic coating with good mechanical stability (Table 3). This coating formed by the fluorinated peptide displays better antiviral activity than the non-fluorinated coating for COVID-19 and T4 bacteriophage, indicating the stabilizing effect of fluorine on the peptides’ supramolecular structure contributes to the formation of a superior antiviral coating. When assembled into a three-layer coating, the antiviral peptides lyse the T4 virus particles by 10^3^ PFU/mL off the glass surface; the non-fluorinated peptide acting as a negative control only knocks down the virus titers by a negligible amount [71]. However, when examining the antiviral activity of the peptides with coronavirus, both the fluorinated and non-fluorinated peptides decrease below the system’s limit of detection, indicating significant viral eradication. With no distinguishable difference between the two treatments, the anti-SARS-CoV-2 activity of these bioactive peptides from its rationally designed physical structure inhibits viral infiltration of cell membranes, rather than from any inductive chemical change from the addition of fluorine. Overall, while fluorination of bioactive peptides may introduce viral killing behavior, the extent of this is heavily dependent on the identity of the pathogenic viral strain.

### 5.2. Fluorinated Antibodies

Antibodies belong to a class of large proteins naturally made by the immune system with the sole purpose to bind to specific markers on cells or tissues [76]. By making use of their antigen specificity, engineered therapeutic antibodies are instrumental in launching pinpointed attacks against specific antigens of interest. Therapeutic antibodies have therefore attracted attention as a viable treatment option for a vast array of disease types, including autoimmune disorders, cancer, viral infections, asthma, poisoning, and substance abuse; however, the chemical design of therapeutic antibodies is tricky, given their short half-life and incredibly specific active site requirements [77]. Fluorination is a potential way of aiding in this design, as its small size could provide better shape complementarity, allowing the antibody to better dock to antigen binding sites. To test the potential of these effects, Barchi et al. have constructed several single-chain antibodies from 2ZNX with either complete or individual replacement of tryptophan residues with 5-fluorotryptophan (^5F^W), designed to bind to antigens from a human erythroleukemia cell line (HEL) (Table 3) [72]. Their design selection is based off the crystal structure of the native antibody bound to the antigen (Figure 14A): because the binding site already has a high presence of hydrophobic tryptophan residues that arrange into a unique shape, the authors replace these residues with fluorinated analogues in their attempt to increase its hydrophobicity while matching the shape of the pocket (Figure 14B). Thus, fluorinated amino acids can be applied to fine-tune properties such as protein folding, proteolytic stability, and protein−protein interactions provided we understand and become able to predict the outcome of a fluorine substitution in this context.

Using surface plasmon resonance (SPR) to derive antigen-binding affinities, the fluorinated variant displayed approximately 31 times weaker antibody-antigen binding compared to the non-fluorinated analogues (Figure 14C). While this weakened binding was not anticipated, it is hypothesized that the electronegative properties of fluorine resonance stabilized tryptophan’s indole ring present in the antibody’s active site, rendering it unreactive. Hence, fluorine’s typically helpful stabilization effect translated negatively as a result of perhaps too much stabilization in what should be a reactive and unstable active site. This counterintuitive effect could shed some insight on why fluorine is not a common technique used in the design of therapeutic antibodies. Another potential reason is that antibodies are already large and relatively stable proteins, so to observe an appreciable stabilizing effect, a high amount of fluorine would have to be added to the molecule, which would in turn negatively affect its solubility. Therefore, for these types of systems, fluorination would primarily be used to enhance binding interactions or pharmacokinetics and pharmacodynamics, which is heavily dependent on the context of the active site.

### 5.3. Collagen Mimetics for Fiber Repair

Applications of collagen mimetic peptides (CMPs) are continuing to advance our understanding of the structure and molecular properties of a collagen triple helix and the interactions of collagen with important molecular ligands for therapeutic use. Utilizing (ProHypGly), the most common triplet in collagen, Raines et al. have shown that mutating this triplet with a fluorinated AA can lead to improved therapeutic behavior [73]. Specifically, mutating (2*S*,4*S*)-4-fluoroproline (flp) to form (flpHypGly)_7_ will form stable triple helices with (ProProGly)_7_ but will not form triple helices with its other fluorinated monomers (Table 3). Because this triplet is therefore monomeric in solution yet has a high ability to anneal with damaged collagen, the Raines group has demonstrated its ability to detect mammalian collagen in rat tails that has suffered burn damage after conjugation to fluorescent dye Cy5. Second harmonic generation (SHG) microscopy, used to quantify the Cy5 fluorescence and act as a proxy for collagen binding, reveal that the localization of ^Cy5^CMP on burned collagen is greater by 25-fold than its fluorescence on intact collagen. Fluorination therefore seems to be a promising technique to engineer CMPs to avoid the formation of homotrimers while being inherently preorganized to adopt the conformation of a collagen strand, leading to enhanced association with damaged collagen duplexes. This annealing could also allow for the delivery of diagnostic or therapeutic agents conjugated to the CMP, expanding their potential into the diagnosis and treatment of fibrotic diseases and wounds [78]. However, damaged collagen in the human body is a complex target, which is complicating the assessment of new fluorinated CMPs’ therapeutic behavior, hence why studies are currently limited in scope.

## 6. Drug Delivery

The bioavailability of drug candidates is often the largest barrier from clinical translation as issues of solubility, aggregation, degradation, cell-membrane penetration, and clearance often necessitate a carrier molecule to protect and escort drug payloads to their targeted sites [79]. Accordingly, selection criteria for drug-delivering protein or peptide materials share many similarities with those employed for therapeutic use including easy administration, good pharmacokinetics and dynamics, and minimization of off-target effects; however, essential characteristics tend to be more flexible since the vehicle does not need to be intrinsically medicinal in nature [80]. This is often preferred so as not to confer unintended compounding effects.

Leveraging their target specificity to certain tissues, antibodies have emerged as some of the first protein drug delivery vehicles. Several different higher-order assemblies including nucleic acid-protein complexes, solid-state nanoparticles, dendrimers, micelles, and emulsions have since emerged striving to raise the drug-to-carrier ratio, accommodate diverse payloads, and cater towards different routes of administration (Table 4, Figure 15).

**Table 4 pharmaceuticals-15-01201-t004:** Summary of fluorinated proteins and peptides constructs alongside their accompanying one letter AA sequences in the order they are discussed for their application as drug delivery vehicles.

Construct Name	Fluorinated Residue(s)	Number of Fluorines	Position of Fluorines	Synthetic Method	Assembly Morphology	Sequence	References
F-TRAP	Leucine (L)	3 per L (12 total)	Methyl group, δ-carbon	Residue-specific ncAA incorporation	Protein micelles	MRGSHHHHHHGSACE**L**AARGDATATATATAACGD**L**APQMLRE**L**QETNAA**L**QDVRE**LL**RQQVKEITF**L**KNTVMESDASG**L**QAARGDATATATATAVDKPIAASAVPGVGVPGVGVPGFGVPGVGVPGVGVPGVGVPGVGVPGFGVPGVGVPGVGVP**L**EGSGTGAK**L**N	[81]
D1	(*R,R*)-β-diaryl (^β^PhPh)	1 per ^β^PhPh (1 total)	Aryl ring, 2-ortho position	Ligation	Fibrous nanotubes	A*^β^Ph**Ph***	[82]
D2	(*S,S*)-β-diaryl (^β^PhPh)	1 per ^β^PhPh (1 total)	Aryl ring, 2-ortho position	Ligation	Fibrous nanotubes	A*^β^Ph**Ph***	[82]
MfeGlyK16	Homoalanine (^h^A)	1 per ^h^A (8 total)	Ethyl group, γ-carbon	SPPS	Hydrogelator Agent	**^h^****A**K**^h^A**K**^h^A**K**^h^A**K **^h^A**K**^h^A**K**^h^A**K**^h^A**K-*BzOH*	[83]
DfeGlyK16	Homoalanine (^h^A)	2 per ^h^A (16 total)	Ethyl group, γ-carbon	SPPS	Hydrogelator Agent	**^h^****A**K**^h^A**K**^h^A**K**^h^A**K **^h^A**K**^h^A**K**^h^A**K**^h^A**K-*BzOH*	[83]
TfeGlyK16	Homoalanine (^h^A)	3 per ^h^A (24 total)	Ethyl group, γ-carbon	SPPS	Hydrogelator Agent	**^h^****A**K**^h^A**K**^h^A**K**^h^A**K **^h^A**K**^h^A**K**^h^A**K**^h^A**K-*BzOH*	[83]
Fmoc-3F-Phe-DAP	Phenylalanine (F)	1 per F (1 total)	Aryl ring, 3-meta position	SPPS	Hydrogelator Agent	*Fmoc*-**F**-*DAP*	[84]
Fmoc-F**^5^**-Phe-DAP	Phenylalanine (F)	5 per F (5 total)	Aryl ring, all positions	SPPS	Hydrogelator Agent	*Fmoc*-**F**-*DAP*	[84]
4-fluorobenzyl-diphenylalanine	Phenylalanine (F)	1 per F (1 total)	Aryl ring, 4-para position	SPPS	Hydrogelator Agent	***Bz***-FF	[85]
No name provided	Phenylalanine (F)	5 per F (15 total)	Aryl ring, all positions	SPPS	Peptide emulsion	**FFF**GGGCCGGKGRGD	[86]

Non-AA components of the vehicle are denoted in italics. *^β^PhPh* refers to a non-canonical diaryl β-amino acid. ^h^A refers to the non-canonical homoalanine or α-aminobutyric acid (Abu). *Fmoc* refers to the common *N*-fluorenylmethoxycarbonyl protecting group. *DAP* refers to a diaminopropane. *Bz* refers to a benzyl endcap. *BzOH* refers to a benzoic acid endcap. Fluorinated residues are denoted in bold to indicate their location in the sequence.

### 6.1. Spherical, Colloidal Fluorinated Peptide and Protein Vehicles

Protein and peptide amphiphiles are widely employed in drug delivery vehicles owing to their self-assembly of ordered complexes in aqueous solution. Containing discrete hydrophobic and hydrophilic segments, these amphiphilic molecules tend to arrange themselves as spherical particles in solution due to the low interfacial energy associated with this morphology that minimizes surface area per unit volume [87]. In aqueous solutions, robust dispersion of amphiphilic molecules typically manifests as assemblies with protective hydrophilic exteriors and hydrophilic interiors, where hydrophobic guest molecules can reside [88]. Micelles and nanoemulsions refer to the simplest form of these biphasic vehicles containing a single monolayer of the emulsifying agent (as opposed to a liposome comprised of an amphiphilic bilayer folded in on itself). The major difference between these species is the presence of a liquid-phase lipophilic core at the center of nanoemulsions that micelles lack [89]. Although expanding the size of the carrier, nanoemulsions tend to be favorable for aerosol or pulmonary applications since the core better facilitates vaporization and encapsulation of lipophilic small molecules [89,90,91]. The following subsections will delve further into two drug delivery vehicles adapting these morphologies constructed from fluorinated protein and peptide amphiphiles.

#### 6.1.1. Fluorinated Protein Micelles

The Montclare lab has developed a protein block copolymer that forms a micelle, named fluorinated thermoresponsive assembled protein (F-TRAP), composed of a coiled-coil domain and two repeats of an intrinsically-disordered elastin-like polypeptide domain fluorinated with trifluoroleucine (TFL) (Figure 15A, Table 4) [81]. The binding capacity of F-TRAP and non-fluorinated TRAP assemblies have been assessed for Dox. Aimed at controlling its release through thermosensitive coacervation of the protein, F-TRAP encapsulates 175% more Dox than the non-fluorinated TRAP. In addition, comparison of the weighted Dox loading reveals 55% greater loading by F-TRAP compared to TRAP, likely due to the increased hydrophobicity of the coiled-coil pore of F-TRAP upon incorporation of TFL. The release of Dox by F-TRAP and its therapeutic efficacy have been assessed in mammalian tumor cells. The cell viability is reduced by both Dox and F-TRAP•Dox compared to F-TRAP alone, but the overall effect of Dox on cell viability is significantly stronger than that of F-TRAP•Dox. While it therefore appears that F-TRAP might not be as effective in releasing its cargo at normothermic temperatures, subjecting the samples to a hyperthermic state triggers the disassembly of F-TRAP, enabling the strongest effect on cell viability. The work here highlights the ability of utilizing fluorine in protein engineering techniques to leverage the conformational flexibility of proteins so that they can easily deliver payload under thermal triggers. 

#### 6.1.2. Fluorinated Peptide Emulsions

Emulsions offer a potentially powerful route to stabilize formulations of active peptides; alternatively, they can also be created using biocompatible, biofunctional, or bioresponsive peptides. Schneider et al. have created a new class of self-assembled peptide-based nanoemulsions with ultrasound (US)-sensitivity, owing to the perfluorocarbon liquid interior of the nanoparticles, via the incorporation of pentafluorophenylalanine (F_F_) (Table 4) [86]. These peptide emulsions, termed ‘peptisomes’, are generated via templated assembly of their peptide at the interface of fluorinated nanodroplets (Figure 15B), with their size precisely controlled as a function of the perfluorocarbon feed. The fluorescent cargo phalloidin can be readily encapsulated within the nano-peptisome carrier during the assembly process with an encapsulation efficiency of 81%; cell binding of the nano-peptisomes, followed by acoustic vaporization, leads to the delivery of this cargo into the cytoplasm of lung carcinoma cells [86]. Using phalloidin as a fluorescent probe for facile cargo tracking following US-mediated delivery, flow cytometry is used to measure the cancer cells’ intracellular fluorescence. In the absence of any US trigger, nano-peptisomes are not able to effectively deliver fluorescent phalloidin, as indicated by the cells’ intracellular fluorescence signals near 0. However, at US intensities >0 W/cm^2^, phalloidin’s intracellular delivery increases as a function of increasing intensity [86].

This successful demonstration was echoed for the delivery of a variety of proteins as well. Perfluorocarbon nanoemulsions able to encapsulate proteins into their fluorous interior to enable their US-mediated cytosolic transduction into cells with spatiotemporal control have been designed [92]. By utilizing a fluorous encapsulation approach enabled through the discovery of fluoro-amphiphiles, hydrophilic loading proteins can be encapsulated, with an encapsulation efficiency of 71%, without altering their structural and functional integrity. Clinical ultrasound imaging modalities have been used to trigger localized injection of GFP payload across the cell membrane and into the cytoplasm. Similarly to the previous study, in the absence of the acoustic trigger, the nanoemulsion remains intact and does not deliver the payload; conversely, its rapid release is observed when the particles are activated with the US trigger. Semiquantitative analysis of intracellular fluorescence also reveals that >150 times more GFP is delivered into cells in response to the US trigger compared to controls sonicated in the presence of the free protein. Importantly, both studies demonstrate the ability of these stimuli-responsive fluorinated peptide emulsions to not only delivery their payload, but to do so by completely disintegrating upon release, demonstrating the groups’ careful attention towards biodegradability in their design.

### 6.2. Fluorinated Fibers

Various molecular forces, some driven by fluorine-mediated interactions, cause peptides to self-assemble in different supramolecular moieties. Specifically, a peptide with an electrostatically charged/hydrophilic head and a hydrophobic tail can self-assemble into spherical particles as described in *Spherical, Colloidal Fluorinated Peptide and Protein Vehicles*, which when elongated can lead to the formation of fibers or nanotubes. Types of fiber-forming materials include α-helical protein-based architectures that form multi-strand, linear coiled-coil assemblies [56], along with flat structures such as tapes or ribbons constructed from *β*-sheets [82]. Increasing concentrations of these proteins and peptides lead to the individual linear strands bundling with one another and turning into more firmly packed fibers [93]. Although not much in vitro or in vivo work has been done with fluorinated peptide and protein fibers, other proof-of-concept studies are showing how fluorination can aid in templating fibril assembly [56,94,95].

Gelmi and coworkers have recently synthesized dipeptides constructed from a canonical alanine joined to a noncanonical diphenyl *β*-amino acid to use in the formation of stable nanotubes [82]. Similar dipeptides have been widely exploited in fiber formation in other previous work, as in aqueous solution it forms channel structures held together by hydrogen bonds along the peptide backbone and π-π interactions between the aromatic rings of the side chains [96,97,98]. The resultant nanoscale tubes are characterized by hydrophobic external walls, while hydrophobic/hydrophilic groups remain exposed on the inner surface. Until the aforementioned study, utilization of fluorine in these peptide nanotubes for drug delivery applications has been limited. To encourage the self-assembly of their nanotubes on a larger scale to form tubular bundles, while minimizing protease degradation, Gelmi et al. have synthesized the (*S,S*) and (*R,R*) dipeptide diastereomers D1 and D2, respectively (Table 4) [82]. The aryl groups present induce π-π interactions and sterically shield the carbon-carbon bond of the *β*-amino acid, leading to resistance from proteolysis. Additionally, the fluorine atom added into the *β*-amino acid residue is proposed to increase lipophilicity, a property that typically aids in cellular uptake. Because the most common transport route of a drug delivery vehicle is through membranes via passive diffusion, vector candidates need to be lipophilic enough to penetrate through the lipid core of membranes without the need for external energy [99]. Furthermore, bundling of nanotube rods is facilitated by in-plane, bifurcated hydrogen bonding between a C-H and the carbonyl oxygen, as well as the aryl fluorine atom. Only the (*R,R*) diastereomer, D2, forms ordered nanotubular (NT) structures in aqueous solution, forming the higher-ordered NT-D2, as the (*S,S*) configuration orients the aryl fluorine away from the C-H bond (Figure 15C). Cytotoxicity and cellular incorporation studies reveal that NT-D2 is not cytotoxic at concentrations up to 50 μm and is successfully incorporated into cells within the cytoplasmic/perinuclear region [82]. The internal hydrophobic nature of these fibers leads to the possibility of a small hydrophobic drug being encapsulated, provided that it fits lengthwise into the fiber’s 4 Å diameter. These findings suggest that fluorinated peptide fibers could serve as scaffolds for novel drug delivery systems.

### 6.3. Fluorinated Protein and Peptide Hydrogels

Peptide hydrogels form when individual, high-aspect-ratio peptides physically or chemically crosslink in aqueous solution [100]. They have the tendency to absorb considerable amounts of water within their network due to a presence of branched polar hydrophilic moieties, while maintaining their crosslinked structure in the swollen state. A pioneer in the field of fluorine chemistry, Koksch and her group have established broad principles governing the use of fluorine in peptide design to drive hydrogel self-assembly [83]. Crafting a library of dipeptides from a cationic scaffold comprised of the non-canonical AA homoalanine (^h^A) also known as α-aminobutyric acid (Abu) and lysine, the group systematically varied the degree of fluorination to directly assess the impact of fluorine-specific interactions on self-assembling gelation behavior. Monofluorinated, difluorinated, and trifluorinated γ-carbon derivatives of ^h^A were synthetically incorporated to form the following 16-mer oligopeptides: MfeGlyK16, DfeGlyK16, and TfeGlyK16, which were all observed to form peptide hydrogels at physiological pH 7.4 (Table 4). Investigating macroscopic mechanical properties, specifically the plateau storage moduli (G_0_) which serves as a proxy of gel crosslinking density, the MfeGlyK16 gel was found to have the lowest *G_0_* value (0.53 Pa) compared to original scaffold constructed from non-fluorinated ^h^A and lysine residues (4.81 Pa), which were both significantly lower than the difluorinated DfeGlyK16 (15.3 Pa) and trifluorinated TfeGlyK16 (670 Pa) [83]. These soft gels all display at least an order of magnitude less than the positive control of an analogous 16-mer comprised of repeating leucine and lysine dipeptides (4869 Pa), which the authors attribute to the higher degree of harmonious hydrophobicity displayed in the non-polar side chains [83]. Thus, the successful addition of fluorine atoms to the scaffold seemed to strengthen hydrogel viscoelastic stability under physiological conditions, but not more than a rationally mutated hydrophobic residue, suggesting the fluorous effect to be less prominent than the hydrophobic effect in the self-assembly of peptide hydrogels.

Many peptide hydrogels also have the ability to increase the dwelling period of drugs due to their mucoadhesive and bioadhesive characteristics, promoting them as suitable nominees for drug carriers [101]. Using the common SPPS protecting group *N*-fluorenylmethoxycarbonyl (Fmoc) conjugated to a phenylalanine residue and a positively-charged diaminopropane (DAP) to improve water solubility, Nilsson et al. created the Fmoc-Phe-DAP scaffold to similarly promote the spontaneous self-assembly of hydrogels (Figure 15D, Table 4) [84]. Hypothesizing that enhancement of π-π stacking interactions between the aromatic Fmoc and Phe would further drive supramolecular hydrogel assembly, they have created a series of three Fmoc-Phe-DAP molecules with varied degrees of fluorinated phenylalanine rings (Table 4). Inversion of the aromatic dipole enables complementary charged interactions between neighboring fluorinated and non-fluorinated aromatics allowing for more direct sandwich stacking and Van der Waals packing of the phenyl groups [84]. While all are able to spontaneously self-assemble in aqueous solutions, the perfluorinated Fmoc-F^5^-Phe-DAP hydrogel exhibits the highest entanglement of peptide chains and the strongest long-term stability over a two-week period, demonstrating its potential as a drug delivery vector for sustained release [84].

Diclofenac, an anti-inflammatory drug was loaded into all three derivatives and their in vitro drug release profiles were collected. The release rate of diclofenac was slowest from the pentafluorinated hydrogel with a value of 1.23 × 10^−13^ m^2^/min, followed by the non-fluorinated Fmoc-Phe-DAP and, via the meta position (3) of the phenyl group, the monofluorinated Fmoc-3F-Phe-DAP with values of 9.54 × 10^−13^ and 9.71 × 10^−13^ m^2^/min, respectively [84]. Presumably, aromatic diclofenac molecules participated in specific π-π interactions via altered quadruple bonding with the aromatic phenyl side chains that was not possible with the non-aromatic analogue. The differences in the amount of released diclofenac could be attributed to differences in affinity to the peptide gelators. Inverting the quadrupole of the aromatic benzyl side chain in the perfluorinated Fmoc-F^5^-Phe-DAP presented more attractive π-π interactions between the peptide and cargo, explaining its slower release profile compared to monofluorinated Fmoc-3F-Phe-DAP and non-fluorinated Fmoc-Phe-DAP.

Similar to fibrous delivery vehicles, proof-of-concepts using fluorinated moieties on peptides for hydrogel assembly have been reported, but few have been tested for drug delivery. For example, Lin et al. have developed an amphiphilic hydrogelator, 4-fluorobenzyl-diphenylalanine, along with the analogue benzyl-diphenylalanine for comparison (Table 4) [85]. Rather than using the bulkier Fmoc, interactions between the smaller perfluorobenzene (PFB) and benzene were exploited for assembly, which rely on alternative, parallel stacking of PFB and the benzyl side chain of L-Phe. 4-fluorobenzyl-diphenylalanine self-assembles into nanofibers, followed by hydrogels, in the range of 1.5 wt% to 5 wt% at a pH of 7.89 [85]. However, benzyl-diphenylalanine is unable to gel at any concentration less than 5 wt%; even at 5 wt%, a large amount of precipitation in the gel solution was observed [85]. This dramatic change between both peptides highlights the strong effect of PFB on self-assembly and hydrogelation, even though the moiety is only monofluorinated. While no drug delivery studies are investigated, a 3-(4,5-dimethylthiazol-2-yl)-2,5-diphenyl-2H-tetrazolium bromide (MTT) cell viability assay has been conducted to assess biocompatibility. Survival ratios of 4-fluorobenzyl-diphenylalanine and benzyl-diphenylalanine are 75% and 45% for PC-3 cells, respectively, indicating that the fluorinated analogue is relatively biocompatible compared to its non-fluorinated counterpart [85]. Both studies reflect the potential of fluorinated moieties as scaffolds for fibril hydrogels with sustained release capabilities; the smaller PFB moiety allows for facile hydrophobic collapse and thus a stronger crosslinked assembly.

## 7. Bioimaging

Bioimaging is a noninvasive process that allows for precise tracking of metabolites used as biomarkers for disease identification, progress, and treatment response. It serves as a window through which organisms and their cells may be observed, their localization and behavior in response to therapeutic treatment that would otherwise be hidden. Despite the advantages bioimaging can offer, improving resolution and specificity could allow us to optimize the clarity in observing specific cells on the molecular and atomic levels [102]. A potential avenue to do so is through fluorinated contrast agents, as fluorine is not endogenously found in high levels, giving it its own unique signature. Transient tracking of certain cells without damaging them is another key challenge in clinical research, encouraging the development of naturally derived contrast agents. Proteins pose the advantage of being genetically encoded and able to become integrated within the cell or organism, in contrast to a dye that can only be labeled from the outside. Performing cell labeling with engineered proteins as the basis of the bioimaging agent therefore reveals exciting possibilities for tracking cells without having to interfere or damage the living tissue, allowing us to study diseases or therapeutics in the host’s natural state and therefore understand diseases better. Taken together, fluorinated proteins have been the center of recent research efforts surrounding common bioimaging techniques, such as Magnetic Resonance Imaging (MRI) and Positron Emission Tomography (PET) (Table 5).

### 7.1. Fluorine-19 Magnetic Resonance Imaging

The low endogenous abundance of fluorine-19 (^19^F) coupled with its favorable +½ spin characteristics have been exploited to improve the therapeutic and diagnostic utility of materials used in photodynamic therapy, MRI and ultrasound. The electronegativity and fluorophilicity of perfluorocarbons, for instance, makes them exceptional passive oxygen carriers across a range of pH and temperature conditions. From this, MRI has seen dramatic improvements in imaging resolution using ^19^F-MRI contrast agents, which take advantage of limited fluorine content in biologic tissues to improve signaling contrast over the typically used ^1^H nuclei abundantly found in biological systems [106]. Similar to therapeutics, it is important for any biological imaging system that the substances used as contrast agents act biorthogonal: they should not alter the properties of cells, cause cytotoxic effects, or change their phenotype or overall behavior. Although a well-thought out design in fluorinated peptides could aid in shielding fluorine’s toxic effects in cells short-term, in regards to clinical translation, it is also important to investigate how these agents are metabolized and cleared from the body (biological half-life) once they leave the cells they initially resided in [107].

The expansion of using pH and Ca^2+^ concentrations as stimuli for ^19^F-MRI stimuli-responsive ‘smart’ contrast agents has largely been a result of Meade et al.’s recent work [103,104]. With regards to pH, they have recently developed peptide amphiphiles with perfluoro endcaps (C7K2, C7E2, C8K2, and C8E2) as responsive MRI contrast agents, with their amphiphilic nature resulting from the hydrophobic perfluorinated tail and hydrophilic AA headgroups on the opposing end (Table 5). These peptides have been designed to optimize pH-responsive ^19^F MRI signals through their use of zwitterionic glutamic acid (E) or lysine (K), whose acidity and basicity enable responsiveness to low and high pH signals, respectively (Figure 16A). Supramolecular aggregates formed by the amphiphiles undergo a morphological change from low-curvature nanoscale ribbons to cylindrical nanofibers as pH is increased or decreased, resulting in enhanced MRI signals. Additionally, shortening the perfluorocarbon tail from C8 to C7 improves the terminal CF_3_ signal to noise ratio (SNR) as there are less rotational degrees of freedom that allow concentration of the signal as a narrow peak [103].

Comparatively, extension of the polar head from E2 to E3 in peptide amphiphile C7E3 allows for more charged sites for the coordination of calcium ions (Ca^2+^) [104]. The authors demonstrate a similar concept in which ^19^F NMR signal intensity is modulated as a function of Ca^2+^ concentration (Figure 16B). These nanostructures demonstrate significant reduction in ^19^F-NMR signal as nanoribbon width increases in response to Ca^2+^, corresponding to ^19^F-MRI image intensity reduction. Their nanostructures respond strongly to Ca^2+^ between 2 and 6 mM, a range encompassing physiological extracellular calcium concentration fluctuations [104]. Their excellent Ca^2+^ sensitivity disappears by an additional glutamic acid residue in their peptide sequence, demonstrating how subtle changes in the chemical structure can have a significant impact on the intensity of ^19^F-NMR signals, rendering them sensitive ^19^F-MRI imaging agents.

In other work, Yuan et al. have developed stimuli-responsive MRI probes in which ^19^F-labeled 3,5-bis(trifluoromethyl)benzoic acid is conjugated to the side chain of lysine in a 6-mer enzyme-cleavable peptide (Figure 17, Table 5) [105]. Through this design, the group have developed a stimuli-responsive method of glutathione (GSH)-controlled assembly and caspase 3/7-controlled disassembly of their nanoparticles that turns the ^19^F-NMR signal “off” and then “on” for the sequential detection of glutathione and caspase 3/7 in vitro and in cells, yielding promising behavior in response to enzyme concentration. Lastly, the F-TRAP protein block copolymer previously described in *Fluorinated Fibers* serves a dual function as a therapy and diagnostic “theranostic” agent (Figure 15B, Table 4) [81]. This assembly results in a drastic depression in spin-spin relaxation (*T*_1_) times and unaffected spin-lattice relaxation (*T*_2_) times. The nearly unchanging *T*_1_ relaxation rates and linearly dependent *T_2_* relaxation rates have allowed for detection via zero echo time ^19^F MRI. The production of a discernible signal with very few fluorinated mutations within this protein construct is significant, holding excitement for what could potentially be done with an even higher fluorine presence. Reflecting on these studies, there is notable promise in utilizing fluorine as ^19^F-MRI contrast agents for a range of biological stimuli, which could provide enhanced imaging resolution in a variety of clinical applications. 

### 7.2. Fluorine-18 Radiotracers in Positron Emission Tomography Imaging

Positron Emission Tomography (PET) is a powerful, minimally-invasive bioimaging technique that can provide real-time molecular, functional, and metabolic information by detecting positron and gamma radiation emitted during decay of radionuclides [108]. Given the cytotoxic, oncogenic, and DNA-damaging nature of radioactivity, PET is often reserved as a secondary diagnostic solution supplementing other imaging techniques, such as correlative computed tomography imaging [109]. Nevertheless, PET remains an expanding diagnostic modality in oncology, neurology, cardiology, and other medicinal disciplines. Clinically, PET can distinguish between benign and malignant tumors, as malignant tumors consume metabolites at a faster rate than benign tumors. The PET principle proves valuable to measure pharmacokinetics and pharmacodynamics of therapeutic candidates and drug delivery vehicles, too.

The bioisosteric nature of fluorine often permits facile installation of ^18^F into large biomolecules without significant alterations to the physiochemical or biological properties of the compound. Traditionally, PET is accomplished through a radiolabeled sugar like ^18^F-fluorodeoxyglucose (^18^F-FDG) whose radioactive signature can be tracked, quantified, and measured along its metabolic journey [109,110]. However, detachment of ^18^F during transit and wide variation in ^18^F-FDG uptake across cell lines and tissue types has contributed to frequent false-positive mislabeling of healthy cells as malignant. To minimize these off-target effects, ^18^F-labeled AA-based radiotracers have been proposed as promising alternatives to glucose-based ones since their uptake by upregulated L-amino acid transporters are linked to more specific cellular activities than generic energy metabolism (Figure 18). Of the various modalities employed, there appears to be a strong preference for ^18^F-fluorinated aromatic AAs with the fluorine selectively substituted into the meta-position, reducing the likelihood of radiolabel detachment during transit.

Designs of radiotracer scaffolds tend to be limited to small molecule metabolites like glucose and AAs. Nevertheless, a smattering of larger ^18^F PET radiotracers built from proteins, including ^18^F fluorinated antibodies and the prostate-specific membrane antigen (^18^F-PMSA) shown in Figure 18, have been successfully synthesized and employed [111,112]. Compared to the other small molecule radiotracers, which rely on the enhanced permeation and retention effect to reach their destination, proteaginous tracers can capitalize on their excellent tissue penetration, robust target specificity, and non-immunogenic or cytotoxic nature. Despite their potential, the number of ^18^F protein tracers remains low due to the complexities associated with radiofluorination. As is the case for the preparation of other radiotracers and radiopharmaceuticals, time spent on synthesis, administration, and circulation must be accounted for in the half-life of fluorine-18 (110 min) [8]. Thus, ^18^F-fluorination is typically reserved as the final synthetic step prior to administration of the radiotracer in vivo. Most of these routes proceed through a top-down approach where ^18^F is introduced through “direct” fast-acting nucleophilic bulk substitutions or “indirect” ligation of ^18^F-fluorinated modalities, such as the previously mentioned ^18^F-labelled AAs or easily-clickable boron-dipyrromethene dyes [113,114]. Several research groups, including the Ritter [8], Vasdev [113], and Sanford [115] groups, have been developing rapid and sophisticated radiofluorination chemistry utilizing organometallic catalysts to expand facile methods to introduce ^18^F to biological scaffolds.

Radiolabeling proteins with ^18^F has also made measurement and understanding of pharmacokinetics and pharmacodynamics of therapeutic species and drug delivery vehicles relatively simplistic since the agent can easily be tracked along its route. Further applications of ^18^F protein labeling beyond the scope of this review extend to the metabolomics and structural proteomics disciplines where the strategy is commonly approach employed to study enzymatic binding pockets, especially with regards to the development of small molecule inhibitors, as well as protein-protein based interactions [116]. With all these advances and developments, the field of ^18^F radiolabeled proteins and AAs provides promise for a bright future where imaging modalities may be integrated in tandem with therapeutic or drug delivery vehicles for a versatile combination of simultaneous therapy and diagnosis.

## 8. Perspective and Conclusions

Fluorinated proteins and peptides with their unique physiochemical properties and unprecedented structural diversity offer promising alternatives to fluoropolymers that have been recently criticized for their biopersistant nature [5]. This rapidly emerging field holds ample opportunities for basic scientists and applied engineers alike to bridge knowledge and technological gaps in pursuit of innovative biomaterials capable of executing a diverse and sophisticated range of biorthogonal functions.

Despite ample understanding of bioisostere principles derived from fluorinating small molecule pharmaceuticals in medicinal chemistry, fluorinating macromolecular structures remains somewhat unpredictable with significant discordance between theory and practice that convolute rational design. For instance, fluorinated AA building blocks appear to have universally lower theoretical helix propensity scores compared to their non-fluorinated hydrophobic counterparts, yet they are empirically observed to stabilize helices to a greater extent when placed in positions buried away from the aqueous interface in the sequence [42]. This fluorous stabilizing trend is similarly observed in the design and maintenance of fluorinated beta-sheets as well as higher order protein structures like fluorinated coiled-coils, beta-barrels, collagen, and block copolymers [33,34,35,36,56,57,58,59,60]. Nevertheless, the result of replacement is dependent on the number of fluorines, the desired position and orientation of the heteroatoms, the size of the protein or peptide scaffold, and the influence of fluorine on surrounding atoms and functional groups.

Applications of these versatile fluorinated materials have primarily been proposed in the biomedical space as fluorination imbues materials with honed target selectivity [1], strengthened binding affinity [2], enriched metabolic stability, and prolonged circulation times [3]. While the introduction of singular fluorine atoms appears to instill these desired properties in large proteins such as antibodies [72], a higher number of fluorinated moieties distributed throughout lengthy sequences tend to be necessary to observe a more pronounced effect. With many applied fluorinated protein materials at the proof-of-concept stage, there is a paucity of downstream metabolic and clearance mechanistic elucidation to thoroughly characterize pharmacokinetic and pharmacodynamic behavior. Although the explanation that protein materials are intrinsically biodegradable and non-toxically cleared has been widely accepted, further investigation into the metabolic and excretion mechanisms is still warranted given the current severity of biopersistent PFAS.

Further expansion of the field is anticipated to result from the perpetually expanding biopharmaceutical arena and the development of new methods like fluorination via post-translational modifications that expand the ways fluorinated materials are synthesized and processed. Evidently, proper understanding of fluorine’s unique physiochemical properties, the variety of techniques to incorporate fluorine, and the intramolecular complexities present in proteinaceous systems are all essential to rationally design and develop the next generation of transformative materials. Efforts in the mentioned directions may evoke paradigms guiding a new era of versatile fluorinated protein and peptide materials.

## Figures and Tables

**Figure 1 pharmaceuticals-15-01201-f001:**
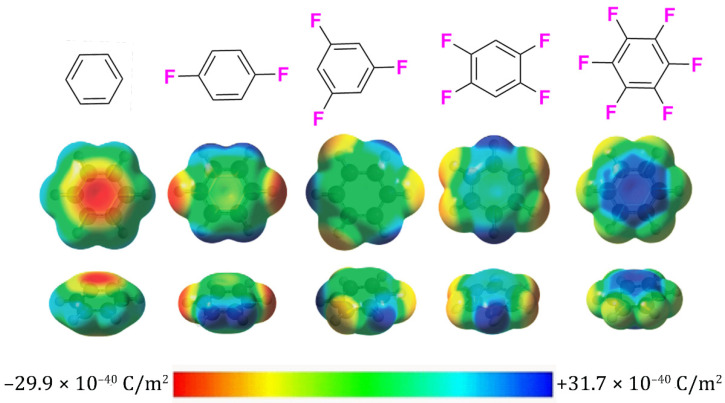
Fluorinated substituent effects on an aromatic π system. Bond-line structures of each system are presented alongside a top and side view of their electrostatic potential maps generated using the Hartree–Fock method in Spartan’18 with the 6-311+G** basis [11]. As more fluorine substituents are appended, the electron-rich π-donor core of the benzene inverts to become an electron-deficient π-acceptor. A rainbow color coded scale ranging from positive blue to negative red depict the spectrum of electrostatic potentials.

**Figure 2 pharmaceuticals-15-01201-f002:**
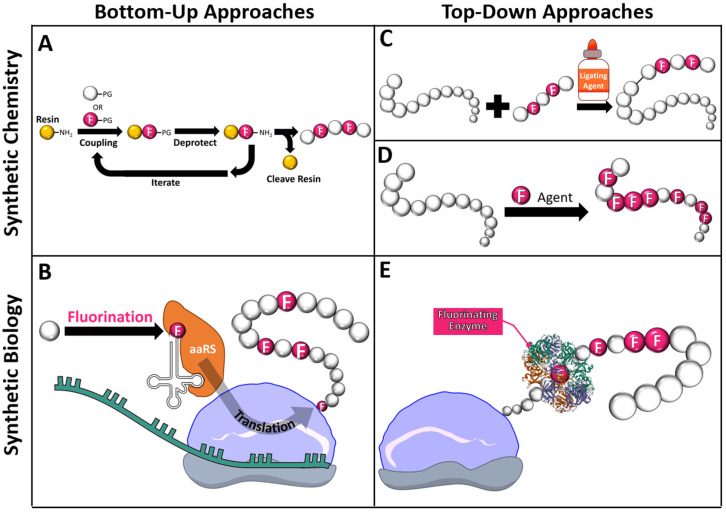
Bottom-up and top-down approaches utilizing various synthetic chemistry and synthetic biology techniques to install fluorine into proteins. Illustrations depicting the generalized process of: (**A**) solid phase peptide synthesis; (**B**) biosynthetic non-canonical amino acid incorporation; (**C**) protein fragment ligation; (**D**) bulk protein fluorination and (**E**) post-translational enzymatic modifications with fluorine. White spheres symbolize building blocks, whereas spheres highlighted in fuchsia and labelled with F represent fluorinated variants. PG signifies protecting group and aaRS indicates aminoacyl tRNA synthetase.

**Figure 3 pharmaceuticals-15-01201-f003:**
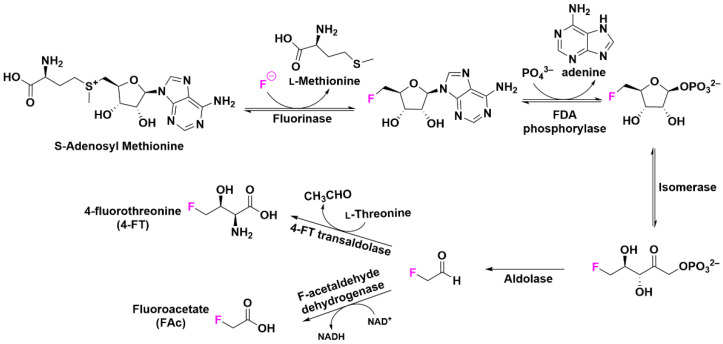
Proposed biosynthetic pathway of to produce FAc and 4-FT in *S. cattleya.* Redrawn from [25]. Copyright 2014 Wang Y. et al. under Creative Commons Attribution License.

**Figure 4 pharmaceuticals-15-01201-f004:**
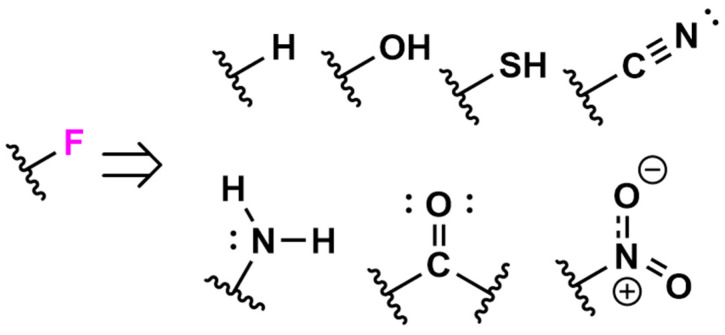
Fluorine as a versatile bioisostere accompanied by the functional groups it has been reported to mimic.

**Figure 5 pharmaceuticals-15-01201-f005:**
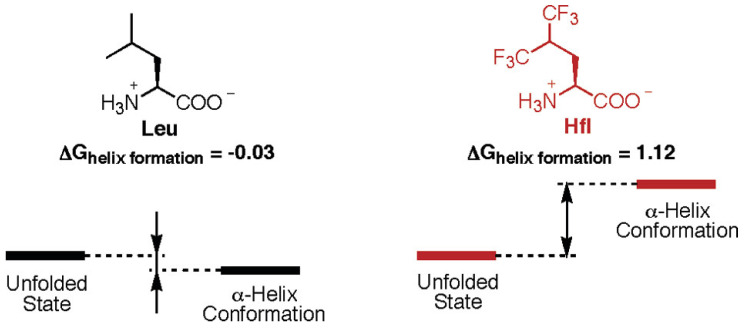
Structures and helical propensities of Leu and Hfl. Reprinted with permission from [32]. Copyright 2006 American Chemical Society.

**Figure 6 pharmaceuticals-15-01201-f006:**
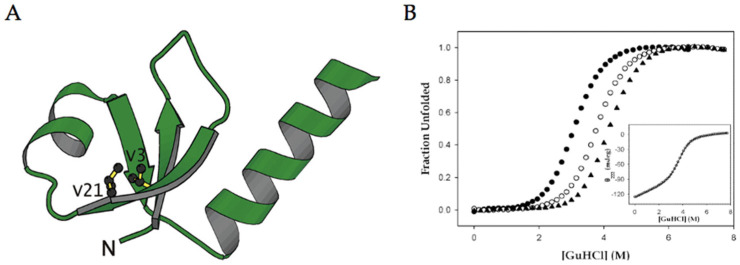
(**A**) Ribbon diagram of NTL9 (PDB code: 1DIV) showing the positions of V3 and V21 which are substituted for 4,4,4-trifluorovaline (tfV3 and tfV21). (**B**) CD monitored equilibrium unfolding of NTL9 (·), tfV3 (○), and tfV21 (▲). Experiments were conducted at pH 5.4, 25 °C, 100 mM NaCl, and 20 mM sodium acetate. Reprinted with permission from [33]. Copyright 2003 American Chemical Society.

**Figure 7 pharmaceuticals-15-01201-f007:**
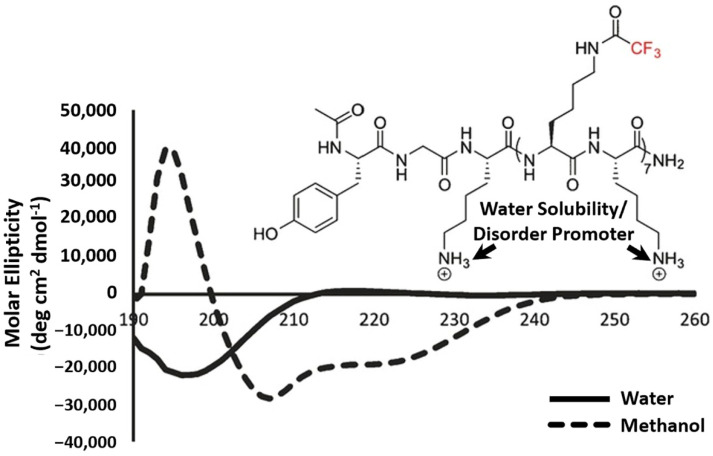
Structure of the intrinsically disordered peptide containing lysine and TFA-lysine accompanied by far-UV circular dichroism spectra in water and in methanol. Adapted with permission from [36]. Copyright 2017 Wiley-VCH Verlag GmbH & Co. KGaA, Weinheim.

**Figure 8 pharmaceuticals-15-01201-f008:**
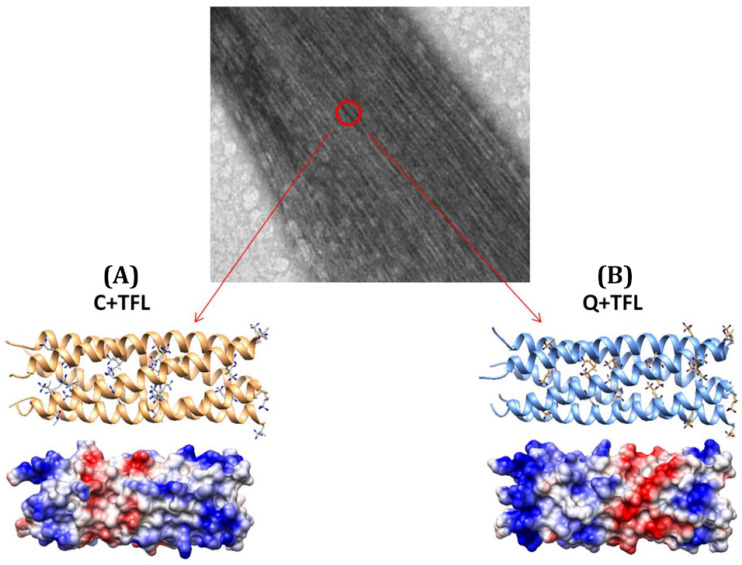
TEM image of a representative coiled-coil fiber formed by (**A**) C+TFL and (**B**) Q+TFL. Ribbon diagrams displaying the spread of TFL side chains are depicted alongside their electrostatic surface profiles where red represents negatively charged regions and blue represents positively charged regions. Reprinted with permission from [56]. Copyright 2015 American Chemical Society.

**Figure 9 pharmaceuticals-15-01201-f009:**
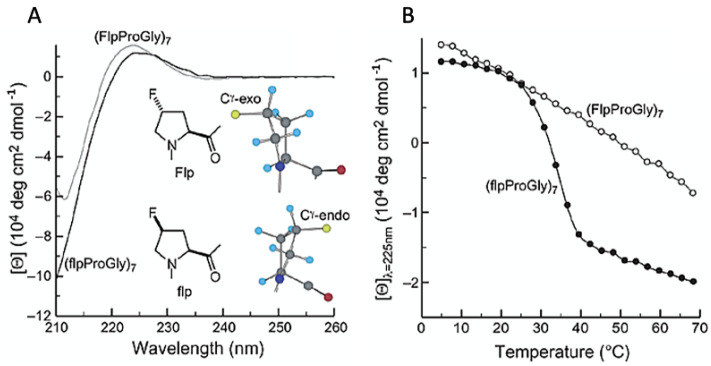
(**A**) Circular dichroism spectra at 5 °C. Inset: Structures of flp and Flp. (**B**) Thermal denaturation curves determined by measuring molar ellipticity at 225 nm as a function of temperature (±1 °C). Reprinted with permission from [63]. Copyright 2003, American Chemical Society.

**Figure 10 pharmaceuticals-15-01201-f010:**
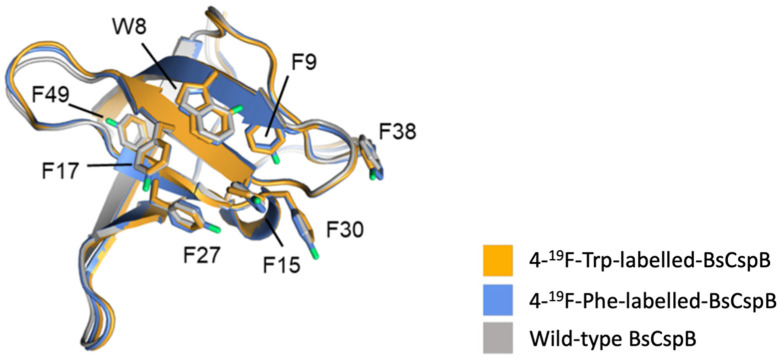
The crystal structures of 4-^19^F-Trp-labelled-BsCspB (orange) and 4-^19^F-Phe-labelled-BsCspB (blue) are shown superimposed onto wild type BsCspB (PDB code 1NMG, grey). The side chains of F-derivatized residues in this study are displayed and labelled according to their sequence numbering. Reprinted with permission from [59].

**Figure 11 pharmaceuticals-15-01201-f011:**
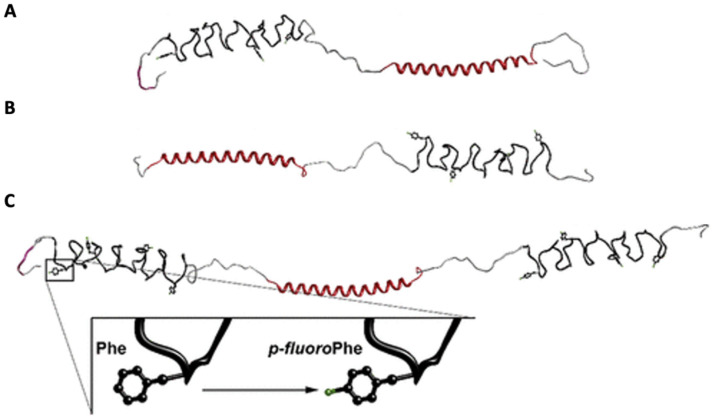
Three block copolymer proteins with residue-specific substitution of pFF. Elastin domains are colored in black and COMP domains are colored in red. Domains are linked by flanker regions containing repeats of AT. (**A**) pFF-EC. (**B**) pFF-CE. (**C**) pFF-ECE. Reprinted with permission from [60]. Copyright 2012 American Chemical Society.

**Figure 12 pharmaceuticals-15-01201-f012:**
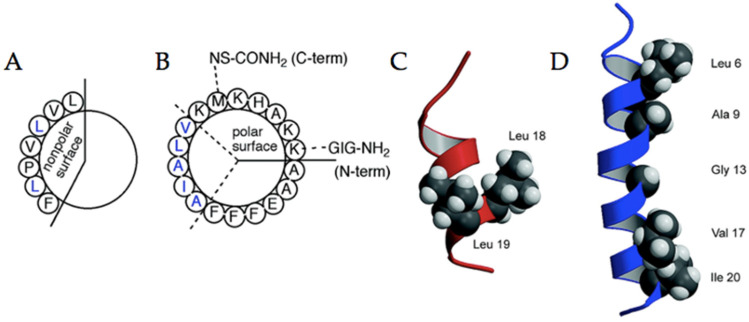
Helical wheel diagrams and sites of fluorination. (**A**) Buforin series: Leu 18 and 19 (blue) were substituted with HFL; (**B**) magainin series sites of fluorination: residues Leu 6 and Ile 20 in M2F2 and Leu 6, Ala 9, Gly 13, Val 17, and Ile 20 in M2F5; (**C**) Model structure of buforin with Leu 18 and 19 shown in space-filling depiction; (**D**) NMR-determined structure of magainin in dodecylphosphocholine micelles. Reprinted with permission from [70]. Copyright 2007 American Chemical Society.

**Figure 13 pharmaceuticals-15-01201-f013:**
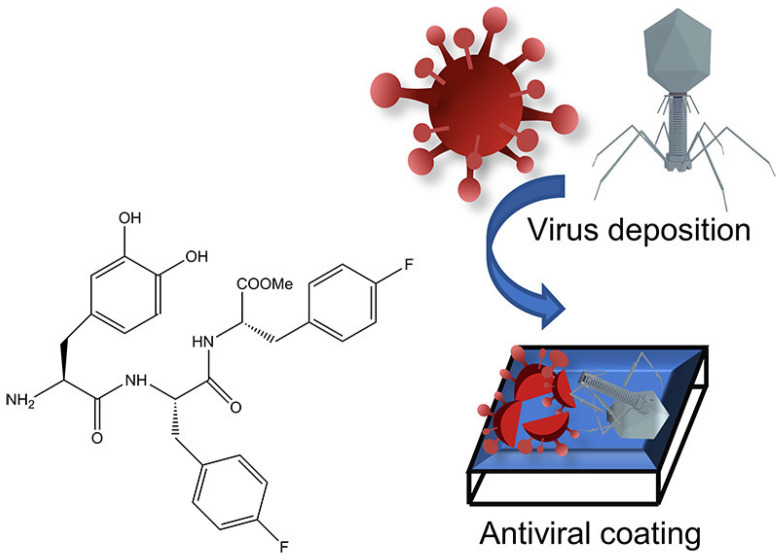
Schematic showing the structure of DOPA-Phe(4F)-Phe(4F)-Ome which utilizes fluorinated phenylalanine residues to kill viruses by forming an antiviral coating when applied to a surface. Reprinted with permission from [71]. Copyright 2021 American Chemical Society.

**Figure 14 pharmaceuticals-15-01201-f014:**
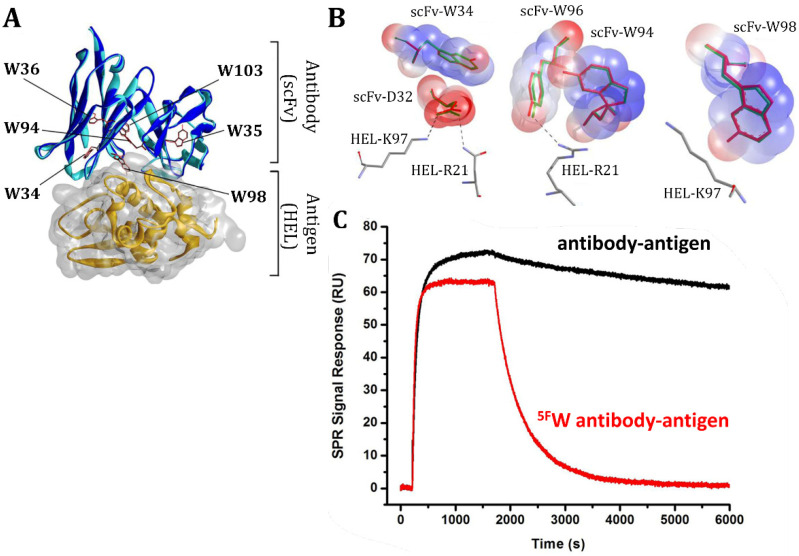
Crystal structures of the antibody-HEL complexes and SPR surface plasmon resonance binding data. (**A**) Ribbon depiction of antibody-HEL complex overlaid with ^5F^W mutations. Antibodies are colored dark blue and cyan, and HEL is colored yellow. Six fluorinated tryptophan residues are annotated. (**B**) Expansions of three key tryptophan residues. Unmodified antibody is colored green and ^5F^W-mutated antibody is colored red. HEL residues are colored gray. Magnification shows any possible steric or electronic influence on the antibody’s interaction with HEL. (**C**) Surface plasmon resonance data demonstrating the impact of complete tryptophan replacement with ^5F^W on antibody-antigen binding over 6000 s. Reprinted with permission from [72]. Copyright 2012 American Chemical Society.

**Figure 15 pharmaceuticals-15-01201-f015:**
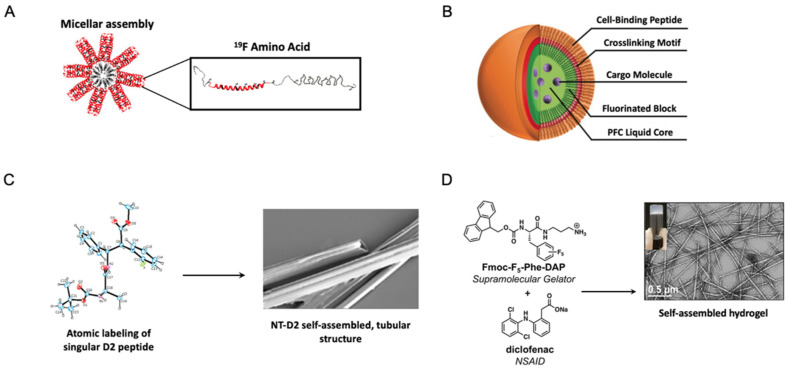
Illustrations of various self-assembled fluorinated protein and peptide drug delivery vehicles. (**A**) Micellar assembly. Adapted with permission from [81]. Copyright 2019 American Chemical Society. (**B**) Nano-peptisome assembly. Adapted with permission from [86]. Copyright 2017 Wiley-VCH Verlag GmbH & Co. KGaA, Weinheim. (**C**) Nanofibers constructed from repeating D2 peptides. Adapted with permission from [82]. Copyright 2015 American Chemistry Society. (**D**) Hydrogel assembled from Fmoc-F_5_-Phe-DAP and loaded with diclofenac, a small molecule non-steroidal, anti-inflammatory drug (NSAID). Adapted with permission from [84]. Copyright 2019 American Chemical Society.

**Figure 16 pharmaceuticals-15-01201-f016:**
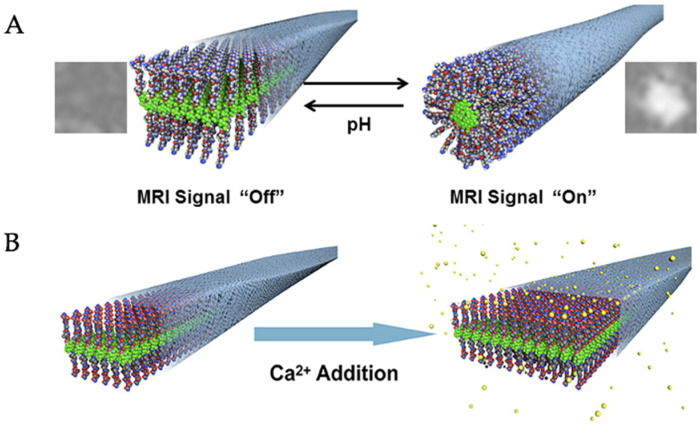
Highly fluorinated peptide amphiphiles that demonstrate changes in morphology and 19F-NMR signal intensity in response to (**A**) pH and (**B**) Ca^2+^ concentration. Figures adapted with permission from [103,104]. Copyright 2017 American Chemical Society.

**Figure 17 pharmaceuticals-15-01201-f017:**
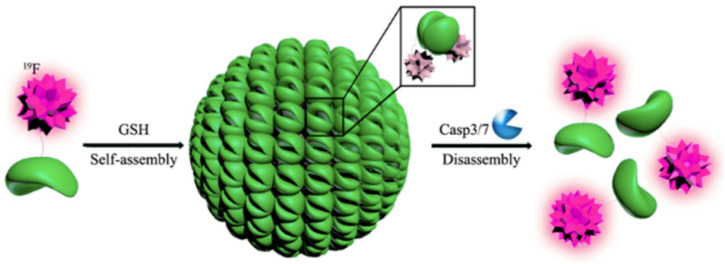
Schematic illustration of GSH-controlled self-assembly to turn ^19^F NMR signals “off” and Casp3/7-controlled disassembly of ^19^F NPs to turn ^19^F NMR signals “on”. Figures adapted with permission [105]. Copyright 2015 American Chemical Society.

**Figure 18 pharmaceuticals-15-01201-f018:**
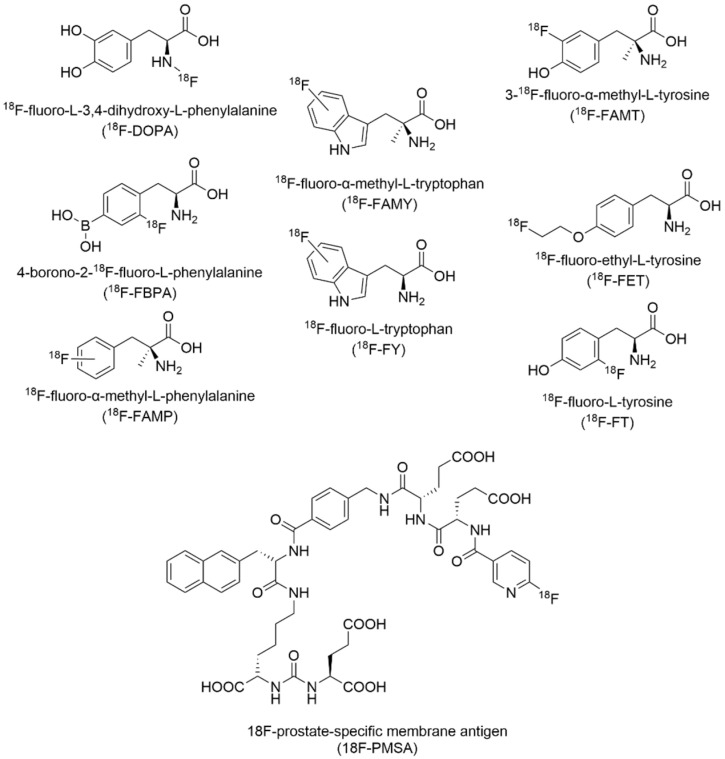
Chemical structures of some ^18^F-L-α-amino acid radiotracers that have been deployed for use as PET diagnostics.

**Table 1 pharmaceuticals-15-01201-t001:** Summary of fluorinated proteins and peptides constructs alongside their accompanying one letter AA sequences in the order they are examined to discern the impact of fluorination on secondary structure.

Construct Name	Fluorinated Residue(s)	Number of Fluorines	Position of Fluorines	Synthetic Method	Secondary Structure	Sequence	References
KHfl	Leucine (L)	6 per L (6 total)	Methyl groups, δ-carbons	SPPS	*α*-helix	YGGKAAAAK**L**AAKAAAAK	[32]
tfV3	Valine (V)	3 per V (3 total)	Methyl group, γ-carbon	SPPS	*β*-sheet	MK**V**IFLKDVKGKGKKGEIK NVADGYANNFLFKQGLAIE ATPANLKALEAQK	[33]
tfV21	Valine (V)	3 per V (3 total)	Methyl group, γ-carbon	SPPS	*β*-sheet	MKVIFLKDVKGKGKKGEIK N**V**ADGYANNFLFKQGLAIE ATPANLKALEAQK	[33]
HH1	Leucine (L)	6 per L (12 total)	Methyl groups, δ-carbons	SPPS	*β*-hairpin	QR**L**TNCCNT**L**EG (cyclic)	[34]
HH2	Leucine (L)	6 per L (12 total)	Methyl groups, δ-carbons	SPPS	*β*-hairpin	**L**@VPAVT**L** (cyclic)	[34]
GB1-QfL	Leucine (L)	4 per L (4 total)	Methyl groups, δ-carbons	SPPS	*β*-sheet	MGYKLALNGKTLKGETT TEAVDAATAEKVFKQYA NDNEGEWAYDDATLKT F**L**VTE	[35]
GB1-Hfl	Leucine (L)	6 per L (6 total)	Methyl groups, δ-carbons	SPPS	*β*-sheet (globular protein)	MGYKLALNGKTLKGETTTE AVDAATAEKVFKQYANDN EGEWAYDDATLKTF**L**VTE	[35]
GB1-Pff	Phenylalanine (F)	5 per F (5 total)	Aryl ring, all positions	SPPS	*β*-sheet (globular protein)	MGYKLALNGKTLKGETTTE AVDAATAEKVFKQYANDN EGEWAYDDATLKTF**F**VTE	[35]
Peptide 1 (n = 7)	Lysine (K)	3 per TFA-modified K (21 total)	Modified acetyl group	SPPS	Intrinsically disordered protein	YGK^TFA^**K**K^TFA^**K**K^TFA^**K**K^TFA^**K**K ^TFA^**K**K^TFA^**K**K^TFA^**K**K	[36]

^TFA^K is used to denote a lysine residue modified with trifluoroacetyl group. @ is used to denote a cyclic non-canonical amino acid 1,2-dihydro-3(6H)-pyridinone used to restrict backbone flexibility and promote *β*-sheet formation. Fluorinated residues are denoted in bold to indicate their location in the sequence.

**Table 2 pharmaceuticals-15-01201-t002:** Summary of fluorinated proteins and peptides constructs alongside their accompanying one letter AA sequences in the order they are examined to discern the impact of fluorination on higher-ordered supersecondary structure.

Construct Name	Fluorinated Residue(s)	Number of Fluorines	Position of Fluorines	Synthetic Method	(Super) Secondary Structure(s)	Sequence	References
Tfl-GCN4-p1d	Leucine (L)	3 per Leucine (12 total)	Methyl group, δ-carbon	Residue-specific ncAA incorporation	*α*-helical coiled-coil	RMKQ**L**EDKVEE**L**LSKNYH**L**ENEVAR**L**KKLVGER	[53]
Hfl-GCN4-p1d	Leucine (L)	6 per Leucine (24 total)	Methyl group, δ-carbon	Residue-specific ncAA incorporation	*α*-helical coiled-coil	RMKQ**L**EDKVEE**L**LSKNYH**L**ENEVAR**L**KKLVGER	[53]
TFL-Protein A1	Leucine (L)	3 per Leucine (24 total)	Methyl group, δ-carbon	Residue-specific ncAA incorporation	*α*-helical coiled-coil	MRGSHHHHHHGSASGD**L**ENE VAQ**L**EREVR S**L**EDEAAE**L**EQKVSR**L**KNEIED**L**KAEIGD**L**NNTSGIRRPAAK**L**N	[54]
HFL-Protein A1	Leucine (L)	6 per Leucine (48 total)	Methyl group, δ-carbon	Residue-specific ncAA incorporation	*α*-helical coiled-coil	MRGSHHHHHHGSASGD**L**ENE VAQ**L**EREVR S**L**EDEAAE**L**EQKVSR**L**KNEIED**L**KAEIGD**L**NNTSGIRRPAAK**L**N	[55]
C+TFL	Leucine (L)	3 per Leucine (21 total)	Methyl group, δ-carbon	Residue-specific ncAAincorporation	*α*-helical coiled-coil	MRGSHHHHHHGSIEGRAPQM**L**RE**L**QETNAA**L**QDVRE**LL**RQQVKEITF**L**KNTSK**L**	[56]
Q+TFL	Leucine (L)	3 per Leucine (21 total)	Methyl group, δ-carbon	Residue-specific ncAA incorporation	*α*-helical coiled-coil	MRGSHHHHHHGSIEGRVKEITF**L**KNTAPQM**L**RE**L**QETNAA**L**QDVRE**LL**RQQSK**L**	[56]
Peptide 2 (GCN4 modified coiled-coil region)	Leucine (L) Valine (V)	3 per L or V (21 total)	Methyl group, δ-carbon	SPPS	*α*-helical coiled-coil	HNRMKQ**L**EDK**V**EE**L**LSKNAS**L**EYE**V**AR**L**KKL**V**GE	[57]
(flpProGly)_7_ collagen triple helix	Proline (P)	1 per P (7 total)	Pyrrolidine ring, 4′-carbon	SPPS	Proline Type II Collagen triple helix	**P**^h^PG**P**^h^PG**P**^h^PG**P**^h^PG**P**^h^PG**P**^h^PG**P**^h^PG	[58]
(GlyProHyp)_10_ collagen model peptide	Phenylalanine (F)	5 per F (10 total)	Aryl ring, all positions	SPPS	Proline Type II Collagen triple helix	**F**GP^h^PGP^h^PGP^h^PGP^h^PGP^h^PGP^h^PGP^h^PGP^h^PGP^h^PGP^h^P**F**	[58]
^19^F-Trp-labelled-BsCspB	Tryptophan (W)	1 per W (1 total)	Aryl ring, 4-, 5-, or 6-carbon	Residue-specific ncAA incorporation	*β*-barrel	MLEGKVK**W**FNSEKGFGFIEVEGQDDVFVHFSAIQGEGFKTLEEGQAVSFEIVEGNRGPQAANVTKEA	[59]
^19^F-Phe-labelled-BsCspB	Phenylalanine (F)	1 per F (7 total)	Aryl ring, 2-ortho, 3-meta, or 4-para-position	Residue-specific ncAA incorporation	*β*-barrel	MLEGKVKW**F**NSEKG**F**G**F**IEVEGQDDV**F**VH**F**SAIQGEG**F**KTLEEGQAVS**F**EIVEGNRGPQAANVTKEA	[59]
pFF-EC	Phenylalanine (F)	1 per F (2 total)	Aryl ring, 4-para-position	Residue-specific ncAA incorporation	Block Copolymer: Disordered elastin-like polypeptide connected to coiled coil	MRGSHHHHHHHGSKPIAASA[(VPGVG)_2_VPG**F**G(VPGVG)_2_]_5_VPLEGSELAATATATATATATAACGDLAPQMLRELQETNAALQDVRELLRQQVKEITFLKNTVMESDASGLQAATATATATATATAVDLQPS	[60]
pFF-CE	Phenylalanine (F)	1 per F (2 total)	Aryl ring, 4-para-position	Residue-specific ncAA incorporation	Block Copolymer: Disordered elastin-like polypeptide connected to coiled coil	MRGSHHHHHHHGSACELAATATATATATATAACGDLAPQMLRELQETNAALQDVRELLRQQVKEIT**F**LKNTVMESDASGLQAATATATATATATAVDKPIAASA[(VPGVG)_2_VPG**F**G(VPGVG)_2_]_5_LEGSGTGAKLN	[60]
pFF-ECE	Phenylalanine (F)	1 per F (3 total)	Aryl ring, 4-para-position	Residue-specific ncAA incorporation	Block Copolymer: Disordered elastin-like polypeptide connected to coiled coil	MRGSHHHHHHHGSKPIAASA[(VPGVG)_2_VPG**F**G(VPGVG)_2_]_5_LEGSELAATATATATATATAACGDLAPQMLRELQETNAALQDVRELLRQQVKEIT**F**LKNTVMESDASGLQAATATATATATATAVDKPIAASA[(VPGVG)_2_VPG**F**G(VPGVG)_2_]_5_LEGSGTGAKLN	[60]

^h^P is used to denote the non-essential hydroxyproline AA. Fluorinated residues are denoted in bold to indicate their location in the sequence.

**Table 5 pharmaceuticals-15-01201-t005:** Summary of fluorinated proteins and peptides constructs alongside their accompanying one letter AA sequences in the order they are discussed for their application as ^19^F MRI agents.

Construct Name	Fluorinated Residue(s)	Number of Fluorines	Position of Fluorines	Synthetic Method	Assembly Morphology	Sequence	References
C7K2	C-terminus	13 per endcap (13 total)	C-terminal endcap	SPPS	Ribbon-like/cylindrical peptide amphiphiles	KKAAVV-*CO-C**F_2_**-C**F_2_**- C**F_2_**-C**F_2_**-C**F_2_**-C**F_3_***	[103,104]
C7E2	C-terminus	11 per endcap (11 total)	C-terminal endcap	SPPS	Ribbon-like/cylindrical peptide amphiphiles	EEAAVV*-CO-C**F2**-C**F2**- C**F2**-C**F2**-C**F2**-C**F3***	[103,104]
C8K2	C-terminus	15 per endcap (15 total)	C-terminal endcap	SPPS	Ribbon-like/cylindrical peptide amphiphiles	KKAAVV*-CO-C**F_2_**-C**F_2_**- C**F_2_**-C**F_2_**-C**F_2_**-C**F_2_**-C**F_3_***	[103]
C8E2	C-terminus	13 per endcap (13 total)	C-terminal endcap	SPPS	Ribbon-like/cylindrical peptide amphiphiles	EEAAVV*-CO-C**F2**-C**F2**- C**F2**-C**F2**-C**F2**-C**F2**-C**F3***	[103]
C7E3	C-terminus	11 per endcap (11 total)	C-terminal endcap	SPPS	Ribbon-like/cylindrical peptide amphiphiles	EEEAAVV-*CO-C**F_2_**-C**F_2_**- C**F_2_**-C**F_2_**-C**F_2_**-C**F_3_***	[103]
Compound 1	Lysine (K)	6 per FMBA-modified K (6 total)	Modified side chain aryl ring (3′ and 5′ positions)	SPPS	Linear Peptide	CDQVD^FMBA^KC	[105]

Non-AA components of the vehicle are denoted in italics. Endcaps are written out in linear atomic format, where C is carbon, O is oxygen, and F is Fluorine. ^FMBA^K is used to denote a lysine residue modified with 3,5-bis(trifluoromethyl)benzoic acid-*N*-hydroxysuccinimide ester. Fluorinated residues are denoted in bold and fuchsia to indicate their location in the sequence.

## Data Availability

Data sharing not applicable.

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
