# Peer review of "Fluorinated Protein and Peptide Materials for Biomedical Applications"

_pharmaceuticals, 2022, doi:10.3390/ph15101201_

Round 1
Reviewer 1 Report
This is a nice review of the role of fluorinated proteins and peptides for biomedical applications. However, there are some significant issues to address. There is not a fluorine chemist as an author and this shows in the Introduction.
The Introduction on the properties of fluorine and its basic chemistry is overly simplistic and in many cases is just borderline correct. Section 1.1 has real issues.
There is no reason to discuss PFAS in this paper as it is not relevant to the topic. Remove this discussion.
Line 41. What do the authors mean by ordinances? This makes no sense in terms of the English language here.
The authors would be better served to describe a few of the key fluorinated drugs that are having a huge impact such as atorvastatin (Lipitor). There are many other examples. One could also mention the fluorinated anesthetics.
What is the value of r for F that is being used on line 74? Is this the van der Waals radius? If so, state that and give the original reference, not just citing the Gouverneur review.
The discussion of the bonding energy in a C-F bond is not correct in terms of the orbitals. One has to discuss the correct hybridization on the C and on F as well as the role of ionic behavior. Also, there should be a discussion of the role of electronegativity of F. What C-F bond energy is being used? The C-F bond energy depends on the number of F atoms attached to the C. Also, give a better reference than the Gouverneur review for the bond energies. The value given is for a CF2 group.
The authors may want to note that the C=C-F group is an isoster for the N-C(O) protein bond.
The authors should discuss that perflurineated carbon chanins with at least 4 carbnon atoms are helical.
At what computational level were the electrostatic maps in Figure 1 generated in Spartan. Spartan has a number of electronic structure methods, so just citing Spartan is useless.
The paragraph starting on line 117 needs revision. I do not think that it is historically accurate and needs to be toned down. The use of the term toxic fluoride should not be used unless detailed references and exact values on how much fluoride is toxic are provided. There is a big difference in wheterh the F is tied up in a compound or as the free fluoride anion. The authors are trying for sensationalism here and it is not appropriate. There are many half truths in this section. Stick to the nice science that they area describing in the later sections. Section 1.1 needs a massive rewrite.
The authors should look at the review chapters by Bruce Smart on C-F chemistry. An example is in the book Organofluorine Chemistry Principles and Commercial Applications eds R. E. Banks, B. E. Smart, J. C. Tatlow. 1994, Springer see chapter on Characteristics of C-F Systems by Bruce E. Smart Smart is an excellent fluorine chemist retired from DuPont who has done excellent surveys on C-F chemistry. See also the initial references in Lemal’s perspective ref. 9 in the current work. There are a number of books on this as well.
I was surprised that the work of Beate Koksch on fluorinated peptides and proteins was not cited. She is a world leader in this area and recently received the ACS Fluorine Chemistry Award of her work in this area.
Also, I was surprised not to see the work of Tobias Ritter or Neil Vasdev among others cited in the work on 18F chemistry for PET imaging. The authors should note that the issue with 18F chemistry is the short half-life which necessitates that any syntheses using it to make compounds for imaging be very fast. Give the 18F half-life.
The main issue with the review is that the Introduction and section 1.1 need real work as well as section 7.2. The rest of the review is very good except for missing the work of Koksch.
Reviewer 2 Report
Manuscript ID: pharmaceuticals-1908889
Title: Fluorinated Protein and Peptide Materials for Biomedical Applications
This review article describes the recent strategies and design principles determining the biochemical synthesis and rational installation of fluorine into protein and peptide sequences for diverse biomedical applications including biomimetic therapeutics, drug delivery vehicles, and bioimaging modalities. The novelty and quality of this review article is good, and the review article summarizes recent publications in this field, including a lot of the researchers’ work, which are good examples for the preparation of various fluorinated protein and peptide materials. The structure of this review article is very well organized. Thus, the review has the potential to provide a valuable asset for the reader. However, to serve this purpose, it should be carefully revised with improved organization, balance, structural detail, reduction in abbreviations or improvement in their use to make everything visually obvious. Some specific suggestions follow:
1. Can the authors comment on the limitations of each one of the strategies presented and future development?
2. The reviewer did not see many data about the biomedical application in this article. Is it possible to improve this part of the discussion? In particular, authors can also cite possible applications in the metabolomics field to a greater detail, if possible.
3. Table 4 is not clear to detailed understand and too tedious to read. Please improve it.
4. Overall, it is recommended for publication in the Pharmaceuticals after the major revisions.
Round 2
Reviewer 2 Report
The MS was properly revised according to the reviewers' suggestions. I recommend its publication without any further changes.